# Dynamic Structure Estimation from Bandit Feedback using Nonvanishing Exponential Sums

**Motoya Ohnishi\***                                                    *mohnishi@cs.washington.edu*
*Paul G. Allen School of Computer Science & Engineering*
*University of Washington*

**Isao Ishikawa\***                                                    *ishikawa.isao.zx@ehime-u.ac.jp*
*Ehime University*
*RIKEN Center for Advanced Intelligence Project*

**Yuko Kuroki**                                                    *yuko.kuroki@centai.eu*
*CENTAI Institute*

**Masahiro Ikeda**                                                    *masahiro.ikeda@riken.jp*
*RIKEN Center for Advanced Intelligence Project*
*Keio University*

**Reviewed on OpenReview:** *https://openreview.net/forum?id=xNkASJLOF6*

## Abstract

This work tackles the dynamic structure estimation problems for periodically behaved discrete dynamical system in the Euclidean space. We assume the observations become sequentially available in a form of bandit feedback contaminated by a sub-Gaussian noise. Under such fairly general assumptions on the noise distribution, we carefully identify a set of recoverable information of periodic structures. Our main results are the (computation and sample) efficient algorithms that exploit asymptotic behaviors of exponential sums to effectively average out the noise effect while preventing the information to be estimated from vanishing. In particular, the novel use of the Weyl sum, a variant of exponential sums, allows us to extract spectrum information for linear systems. We provide sample complexity bounds for our algorithms, and we experimentally validate our theoretical claims on simulations of toy examples, including Cellular Automata.

## 1 Introduction

System identification has been of great interest in controls, economics, and statistical machine learning (cf. Tsiamis & Pappas (2019); Tsiamis et al. (2020); Lale et al. (2020); Lee (2022); Kakade et al. (2020); Ohnishi et al. (2024); Mania et al. (2022); Simchowitz & Foster (2020); Curi et al. (2020); Hazan et al. (2018); Simchowitz et al. (2019); Lee & Zhang (2020)). In particular, estimations of periodic information, including eigenstructures for linear systems, under noisy and partially observable environments, are essential to a variety of applications such as biological data analysis (e.g., Hughes et al. (2017); Sokolove & Bushell (1978); Zielinski et al. (2014); also see Furusawa & Kaneko (2012) for how gene oscillation affects differentiation of cells), earthquake analysis (e.g., Rathje et al. (1998); Sabetta & Pugliese (1996); Wolfe (2006); see Allen & Kanamori (2003) for the connections of the frequencies and magnitude of earthquakes), chemical/asteroseismic analysis (e.g., Aerts et al. (2018)), and communication and information systems (e.g., Couillet & Debbah (2011); Derevyanko et al. (2016)), just to name a few.

---

\* equal contributions

In this paper, we focus on the periodic structure estimation problem for *nearly* periodically behaved discrete dynamical systems (cf. Arnold (1998); specifically, we mainly study theoretical aspects of recovering structural information under a novel set of model assumptions. We allow systems that are not *exactly* periodic with sequentially available bandit feedback. Due to the presence of noise and partial observability, our problem setups do not permit the recovery of the full set of period/eigenvalues information in general; as such we particularly ask the following question: *what subset of information on dynamic structures can be statistically efficiently estimated?* This work successfully answers this question by identifying and mathematically defining recoverable information, and proposes algorithms for efficiently extracting such information.

The technical novelty of our approach is highlighted by the careful adoption of the asymptotic bounds on the exponential sums that effectively cancel out noise effects while preserving the information to be estimated. When the dynamics is driven by a linear system, the use of the Weyl sum Weyl (1916), a variant of exponential sums, enables us to extract more detailed information. To our knowledge, this is the first attempt of employing asymptotic results of the Weyl sum for statistical estimation problems, and further studies on the relations between statistical estimation theory and exponential sums (or even other number theoretical results) are of independent interests.

With this summary of our work in place, we present our dynamical system model below followed by a brief introduction of the properties of the exponential sums used in this work.

**Dynamic structure in bandit feedback.** We define $D \subset \mathbb{R}^d$ as a (finite or infinite) collection of arms to be pulled. Let $(\eta_t)_{t=1}^\infty$ be a noise sequence. Let $\Theta \subset \mathbb{R}^d$ be a set of latent parameters. We assume that there exists a *dynamical system* $f$ on $\Theta$, equivalently, a map $f : \Theta \to \Theta$. At each time step $t \in \{1, 2, \ldots\}$, a learner pulls an arm $x_t \in D$ and observes a reward

$$r_t(x_t) := f^t(\theta)^\top x_t + \eta_t,$$

for some $\theta \in \Theta$. In other words, the hidden parameters for the rewards may vary over time but follow only a rule $f$ with initial value $\theta$. The function $r_t$ could be viewed as the specific instance of partial observation (cf. Ljung (2010)).

**Brief overview of the properties of the exponential sums used in this work.** Exponential sums, also known as trigonometric sums, have developed as a significant area of study in number theory, employing various methods from analytical and algebraic number theory (see Arkhipov et al. (2004) for an overview). An exponential sum consists of a finite sum of complex numbers, each with an absolute value of one, and its absolute value can trivially be bounded by the number of terms in the sum. However, due to the cancellation among terms, nontrivial upper and lower bounds can sometimes be established. Several classes of exponential sums with such nontrivial bounds are known, and in this study, we apply bounds from a class known as Weyl sums to extract information from dynamical systems using bandit feedback. In the mathematical community, these bounds are valuable not only in themselves but also for applications in fields like analysis within mathematics (Katz, 1990). This research represents an application of these bounds in the context of machine learning (statistical learning/estimation problem) and is cast as one of the first attempts to open the applications of number theoretic results to learning theory.

In fact, even the standard results utilized in discrete Fourier transform can be leveraged, with their (non)asymptotic properties, to provably handle noisy observations to extract (nearly) periodic information. This process is illustrated in Figure 1 (the one for period estimation); where we show that one can effectively average out the noise through the exponential sum (which is denoted by $\mathcal{R}$ in Figure 1) while preventing the correct period estimation from vanishing. By multiplying the (scalar) observations by certain exponential value, the bounds of the Weyl sum (which is denoted by $\mathcal{W}$ in Figure 1) can be applied to guarantee the survival of desirable target eigenvalue information; this is illustrated in Figure 1 (the one for eigenstructure estimation).

**Our contributions.** The contributions of this work are three folds: First, we mathematically identify and define a recoverable set of periodic/eigenvalues information when the observations are available in a form of bandit feedback. The feedback is contaminated by a sub-Gaussian noise, which is more general than

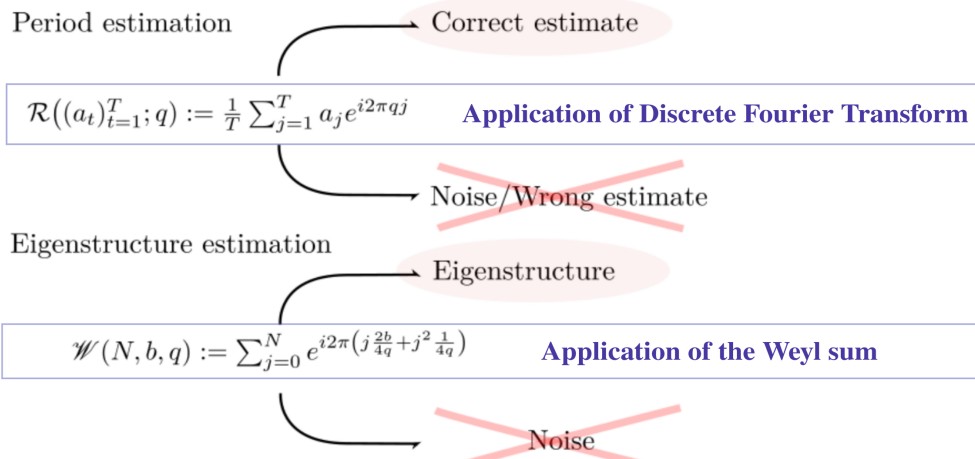

**Figure 1:** Overview of how the exponential sum techniques are used in the work. For (nearly) period estimation problem, it is an application of discrete Fourier transform in our statistical settings, which ensures that the correct estimate remains while noise effect or wrong estimates are properly suppressed. When the system follows linear dynamics in addition to the (nearly) periodicity, our application of the Weyl sum, a variant of exponential sums, preserves some set of the eigenstructure information.

those usually considered in system identification work. Second, we present provably correct algorithms for efficiently estimating such information; this constitutes the first attempt of adopting asymptotic results on the Weyl sum. Lastly, we implemented our algorithms for toy and simulated examples to experimentally validate our claims.

**Summary of technical contributions and challenges.** Here, we showcase a summary of technical contributions and challenges to be overcome:

- Exploiting the exponential sum techniques (including the discrete Fourier transform) studied in number theory community as a *filtering* within the context of statistical estimation problems.

- Adapt the Weyl sum technique to deal with eigenvalues by extending it to matrix sum.

- Defining novel mathematical concepts: *(aliquot) nearly period* and $(\theta, k)$-*distinct eigenvalues*.

- Concurrently applying a set of filterings to statistically identify correct periodic structure (we call this as *concurrent application of filterings* in Algorithm 1).

- System recovery through *maintenance of isomorphic structure* in observation (we call it as *isomorphicity maintenance lemma*; see Lemma C.1).

**Notations.** Throughout this paper, $\mathbb{R}$, $\mathbb{R}_{\geq 0}$, $\mathbb{N}$, $\mathbb{Z}_{>0}$, $\mathbb{Q}$, $\mathbb{Q}_{>0}$, and $\mathbb{C}$ denote the set of the real numbers, the nonnegative real numbers, the natural numbers ($\{0, 1, 2, \dots\}$), the positive integers, the rational numbers, the positive rational numbers, and the complex numbers, respectively. Also, $[T] := \{1, 2, \dots T\}$ for $T \in \mathbb{Z}_{>0}$. The Euclidean norm is given by $\|x\|_{\mathbb{R}^d} = \sqrt{\langle x, x \rangle_{\mathbb{R}^d}} = \sqrt{x^\top x}$ for $x \in \mathbb{R}^d$, where $(\cdot)^\top$ stands for transposition. $\|M\|$ and $\|M\|_F$ are the spectral norm and Frobenius norm of a matrix $M$ respectively, and $\mathscr{I}(M)$ and $\mathscr{N}(M)$ are the image space and the null space of $M$, respectively. If $a$ is a divisor of $b$, it is denoted by $a|b$. The floor and the ceiling of a real number $a$ is denoted by $\lfloor a \rfloor$ and $\lceil a \rceil$, respectively. Finally, the least common multiple and the greatest common divisor of a set $\mathcal{L}$ of positive integers are denoted by $\text{lcm}(\mathcal{L})$ and $\text{gcd}(\mathcal{L})$, respectively.

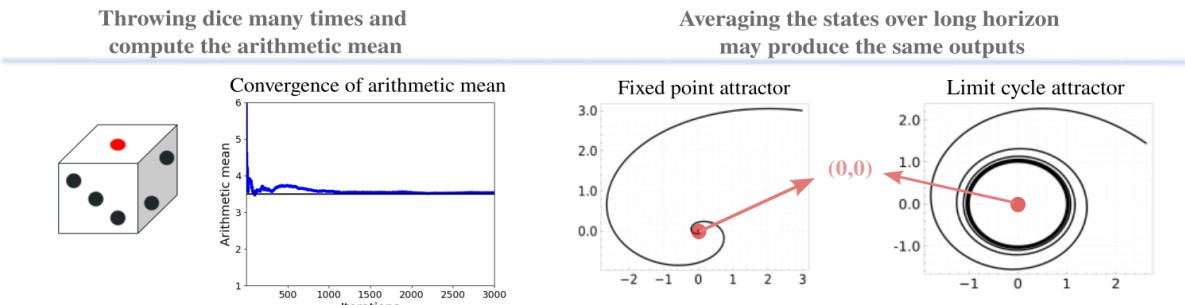

**Figure 2:** Illustration of loss of dynamic information through concentration of measure. Left: An example of the law of large numbers showing the sample mean of the numbers given by throwing dice many times will converge towards the expected value as the number of throws increases. Right: An intuitive understanding of how the concentration of measures or process of averaging out the observations may erase not only the noise effect but also the dynamics information; in this case, the dynamics on the left side is a fixed point attractor and the right one is a limit cycle attractor, both of which may return $(0,0)$ when averaging the states over long time.

## 2 Motivations

As our main body of the work is largely theoretical with a new problem formulation, this section is intended to articulate the motivations behind this work from scientific (theoretical) and practical perspectives followed by the combined aspects for our formulation.

### 2.1 Scientific (theoretical) motivation

When dealing with noisy observation, it is advised to use concentrations of measure to extract the true target value by averaging out the noise effect over sufficiently many samples. On the other hand, when the samples are generated by a dynamical system, averaging over the samples will have an effect of averaging over time, which could lose some information on the evolution of dynamics.

**Dynamic information recovery from noisy observation.** Consider for example a fixed point attractor and a limit cycle attractor dynamics as depicted in Figure 2; when averaging over samples generated by sufficiently long time horizon, both cases would output values that are close to $(0,0)$ which ambiguates the dynamical properties. In particular, properties of long-time average of the dynamics have been studied in ergodic theory (cf. Walters (2000)), and studying dynamical systems from both time and measure is of independent interest.

**Dynamic information recovery from scalar observations.** Also, linear bandit feedback system is a special case of partially observable systems, and specifically considers scalar (noisy) observations.

For reconstruction of attractor dynamics from noiseless scalar observations, as will be mentioned in Section 3, Takens' theorem (or delay embedding theorem; Takens (1981)) is famous, where an observation function is used to construct the embedding. It identifies the dynamical system properties that are preserved through the reconstruction as well.

If the noise is also present, the problems become extensively complicated and some attempts on the analysis have been made (Casdagli et al., 1991); which discusses the trade-off between the distortion and estimation errors.

### 2.2 Practical motivation

Practically, periodic information estimation is of great interest as we have mentioned in Section 1; in addition, eigenvalue estimation problems are particularly important for analyzing linear systems.

**Physical constant estimation.** Some of the important linear systems include electric circuits and vibration (or oscillating) systems which are typically used to model an effect of earthquakes over buildings. Eigenvalue estimation problems in such cases reduce to the estimation of physical constants. Also, they have been studied for channel estimation (Van De Beek et al., 1995) as well.

**Dynamic mode decomposition.** At the same time, analyzing the *modes* of a target dynamical system by the technique named (extended) dynamic mode decomposition ((E)DMD; cf. Tu (2013); Kutz et al. (2016); Williams et al. (2015)) has been extensively researched. The results give spectrum information about the dynamics.

**Attractor reconstruction.** With Takens' theorem, obtaining dynamic structure information of attractors in a finite-dimensional space from a finite number of scalar observations is useful in practice as well; see (Bakarji et al., 2022) for an application in deep autoencoder. Our problems are not dealing with general nonlinear attractor dynamics; however, the use of exponential sums in the estimation process would become a foundation for the future applications for modeling target dynamics.

**Communication capacity.** There has been an interest of studying novel paradigms for coding, transmitting, and processing information sent through optical communication systems (Turitsyn et al., 2017); when encoding information in (nearly) periodic signals to transmit, computing communication capacity that properly takes communication errors into account is of fundamental interest. If the observation is made as a mix of the outputs of multiple communication channels, the problem becomes relevant to bandit feedback as well.

**Bandit feedback − single-pixel camera.** As widely studied in a context of compressed sensing (cf. Donoho (2006); Eldar & Kutyniok (2012); Candes et al. (2006)), reconstruction of the target image from the single-pixel camera (e.g., Duarte et al. (2008)) observations relies on a random observation matrix (or vector) to obtain a single pixel value for each observation. It is advantageous to concentrate light to one pixel for the cases where the light intensity is very low. Bandit feedback naturally coincides with a random observation matrix that the user can design. The observation through a random matrix coupled with compressed sensing has been studied for channel estimation, magnetic resonance imaging, visualization of black holes, and vibration analysis as well.

**Bandit feedback − recommendation system.** Reward maximization problems from bandit feedback have been applied to recommendation system (e.g., Li et al. (2010)) for example, where the observations are in scalar form. Other applications of bandit problems include healthcare, dynamic pricing, dialogue systems and telecommunications. Although our problem formulation is not meant to maximize any reward, it is seamlessly connected to the bandit problems as described below.

**Connection to reward maximization problems.** As we will briefly describe in Appendix E, thanks to the problem formulations in this work, one could use our estimation algorithms for downstream bandit problems (i.e., reward maximization problem with bandit feedback). A naive approach is an explore-then-commit type algorithm (cf. Robbins (1952); Anscombe (1963)) where the "explore" stage utilizes our estimation mechanism to identify the dynamics structure behind the system parameter. Devising efficient dynamic structure estimation algorithms for other types of information than the periodicity and eigenstructure under bandit feedback scenarios may also be an interesting direction of research.

## 2.3 Combined aspects

In our work, we essentially consider periodic (and eigenstructure) information recovery from observation data, which as mentioned is crucial in some application domains. In particular, our formulation cares about the situation where the observation is made by a bandit feedback which produces a sequence of scalar observations. We mention that the bandit feedback is essentially just a (noisy) scalar observation where (possibly random) observation vectors are selected by users.

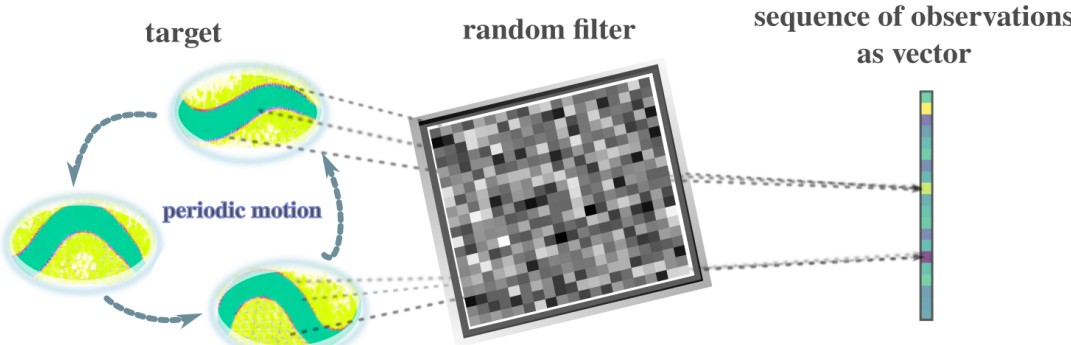

**Figure 3:** Illustration of the example practical case where our techniques may be applied; in this example, the target object is following nearly periodic motion, and requires concentration of light to be captured. A random filter could be employed and we observe a single value sequentially to predict its periodic information.

One possible example scenario in which bandit feedback and dynamic structure estimation meet is illustrated in Figure 3 where the target system has some periodicity and the noisy observation is made through a single-pixel camera equipped with a random filter to be used to determine the periodic structure of the target system.

Also, as mentioned, we emphasize again that reconstruction of dynamical systems characteristics such as their attractor dynamics through (noisy, user-selected) scalar observation is an important area of research; we believe the use of exponential sums in the reconstruction of dynamics opens up further research directions and applications in a similar manner to Takens' theorem (see Section 3 as well).

## 3 Related work

As we have seen in Section 2, our broad motivation stems from the reconstruction of dynamical system information from noisy and partial observations. The most relevant lines of work to this problem are (1) that of Takens (Takens, 1981) and its extensions, and (2) system identifications for partially observed dynamical systems.

While the former studies more general nonlinear attractor dynamics from a sequence of scalar observations made by certain observation function having desirable properties, it is originally for the noiseless case. Although there exist some attempts on extending it to noisy situations (e.g., Casdagli et al. (1991)), provable estimation of some dynamic structure information with sample complexity guarantees (or nonasymptotic result) under fairly general noise case is not elaborated. On the other hand, our work may be viewed as a special instance of system identifications for partially observed dynamical systems. Existing works for sample complexity analysis of partially observed linear systems (e.g., Menda et al. (2020); Hazan et al. (2018); Simchowitz et al. (2019); Lee & Zhang (2020); Tsiamis & Pappas (2019); Tsiamis et al. (2020); Lale et al. (2020); Lee (2022); Adams et al. (2021); Bhouri & Perdikaris (2021); Ouala et al. (2020); Uy & Peherstorfer (2021); Subramanian et al. (2022); Bennett & Kallus (2021); Lee et al. (2020)) typically consider additive Gaussian noise and make controllability and/or observability assumptions (for autonomous case, transition with Gaussian noise with positive definite covariance is required). While (Mhammedi et al., 2020) considers nonlinear observation, it still assumes Gaussian noise and controllability. The work (Hazan et al., 2018) considers adversarial noise but with limited budget; we mention its *wave-filtering* approach is interesting and our use of exponential sums could also be viewed as *filtering*. Also, the work (Simchowitz et al., 2019) considers control inputs and bounded semi-adversarial noise, which is another set of strong assumptions.

Our work is based on the different set of assumptions and aims at estimating a specific set of information; i.e., while we restrict ourselves to the case of (nearly) periodic systems and allow observations in the form of bandit feedback, our goal is to estimate (nearly) periodic structure of the target system. In this regard, we are not proposing "better" algorithms than the existing lines of work but are considering different problem

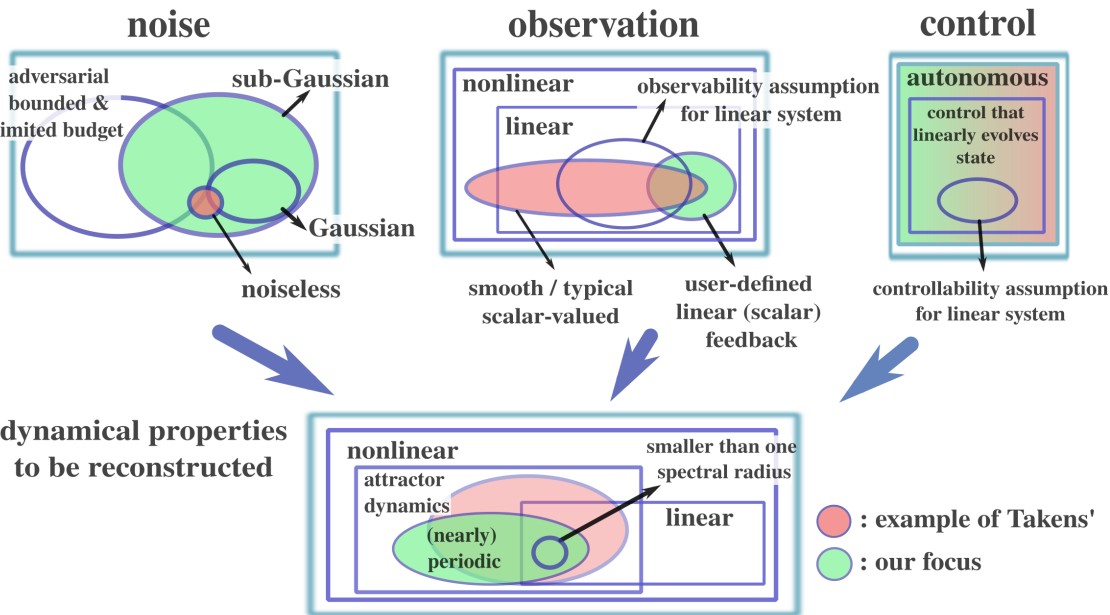

**Figure 4:** Illustration of the position of our work as a study of reconstruction of dynamical system properties. We assume sub-Gaussian noise, bandit feedback (user-defined linear feedback contaminated by noise), autonomous system, and (nearly) periodicity which is the information to be extracted. To our knowledge, our assumptions do not subsume or are subsumed by those of other work as a problem of provably estimating periodic structure. In comparison to existing system identification techniques for partially observed dynamical systems, we *do not* assume Gaussian (or bounded) noise, limited budget for adversarial noise, controllability, observability or noisy transition, while we assume bandit feedback to identify a recoverable set of dynamic structure information.

setups that we believe are well motivated from both theoretical and practical perspectives (see Section 2). We mention that there exist many period estimation methods (e.g., Moore et al. (2014); Tenneti & Vaidyanathan (2015)), and in the case of zero noise, this becomes a trivial problem. For intuitive understandings of how our work may be placed in the literature, we show the illustration in Figure 4.

We also mention that our model of bandit feedback is commonly studied within stochastic linear bandit literature (cf. Abe & Long (1999); Auer (2003); Dani et al. (2008); Abbasi-Yadkori et al. (2011)). Also, as we consider the dynamically changing system states (or *reward vectors*), it is closely related to adversarial bandit problems (e.g., Bubeck & Cesa-Bianchi (2012); Hazan (2016)). Recently, some studies on non-stationary rewards have been made (cf. Auer et al. (2019); Besbes et al. (2014); Chen et al. (2019); Cheung et al. (2022); Luo et al. (2018); Russac et al. (2019); Trovo et al. (2020); Wu et al. (2018)) although they do not deal with periodically behaved dynamical system properly (see discussions in (Cai et al., 2021) as well). For discrete action settings, (Oh et al., 2019) proposed the periodic bandit, which aims at minimizing the total regret. Also, if the period is known, Gaussian process bandit for periodic reward functions was proposed (Cai et al., 2021) under Gaussian noise assumption. While our results could be extended to the regret minimization problems by employing our algorithms for estimating the periodic information before committing to arms in a certain way, we emphasize that our primary goal is to estimate such periodic information in provably efficient ways. We thus mention that our work is orthogonal to the recent studies on regret minimization problems for non-stationary environments (or in particular, periodic/seasonal environments). Refer to (Lattimore & Szepesvári, 2020) for bandit algorithms that are not covered here.

## 4 Problem definition

In this section, we describe our problem setting. In particular, we introduce some definitions on the properties of dynamical system.

### 4.1 Nearly periodic sequence

First, we define a general notion of nearly periodic sequence:

**Definition 4.1** (Nearly periodic sequence). Let $(\mathcal{X}, d)$ be a metric space. Let $\mu \geq 0$ and let $L \in \mathbb{Z}_{>0}$. We say a sequence $(y_t)_{t=1}^{\infty} \subset \mathcal{X}$ is $\mu$-*nearly periodic* of length $L$ if $d(y_{s+Lt}, y_s) < \mu$ for any $s, t \in \mathbb{Z}_{>0}$. We also call $L$ the length of the $\mu$-nearly period.

Intuitively, there exist $L$ balls of diameter $\mu$ in $\mathcal{X}$ and the sequence $y_1, y_2, \ldots$ moves in the balls in order if $(y_t)_{t=1}^{\infty}$ is $\mu$-nearly periodic of length $L$. Obviously, nearly periodic sequence of length $L$ is also nearly periodic sequence of length $mL$ for any $m \in \mathbb{Z}_{>0}$. We say a sequence is periodic if it is 0-nearly periodic. We introduce a notion of aliquot nearly period to treat estimation problems of period:

**Definition 4.2** (Aliquot nearly period). Let $(\mathcal{X}, d)$ be a metric space. Let $\rho > 0$ and $\lambda \geq 1$. Assume a sequence $\{y_t\}_{t=1}^{\infty} \subset \mathcal{X}$ is $\mu$-*nearly periodic* of length $L$ for some $\mu \geq 0$ and $L \in \mathbb{Z}_{>0}$. A positive integer $\ell$ is a $(\rho, \lambda)$-*aliquot nearly period* $((\rho, \lambda)$-*anp*$)$ of $(y_t)_{t=1}^{\infty}$ if $\ell | L$ and the sequence $(y_t)_{t=1}^{\infty}$ is $(\rho + 2\lambda\mu)$-nearly periodic.

We may identify the $(\rho, \lambda)$-anp with a $2\lambda\mu$-nearly period under an error margin $\rho$. When we estimate the length $L$ of the nearly period of unknown sequence $(y_t)_t$, we sometimes cannot determine the $L$ itself, but an aliquot nearly period.

**Example 4.1.** A trajectory of finite dynamical system is always periodic and it is the most simple but important example of (nearly) periodic sequence. We also emphasize that if we know the upper bound of the number of underlying space, the period is bounded above by the upper bound as well. These facts are summarized in Proposition 4.1. The cellular automata on finite cells is a specific example of finite dynamical systems. We will treat LifeGame Conway et al. (1970), a special cellular automata, in our simulation experiment (see Section 6).

**Proposition 4.1.** *Let $f : \Theta \to \Theta$ be a map on a set $\Theta$. If $|\Theta| < \infty$, then for any $t \geq |\Theta|$ and $\theta \in \Theta$, $f^{t+L}(\theta) = f^t(\theta)$ for some $1 \leq L \leq |\Theta|$.*

*Proof.* Since $|\{\theta, f(\theta), \ldots, f^{|\Theta|}(\theta)\}| > |\Theta|$, there exist $0 \leq i < j \leq |\Theta|$ such that $f^i(\theta) = f^j(\theta)$ by the pigeon hole principal. Thus, $f^t(\theta) = f^{j-i+t}(\theta)$ for all $t \geq |\Theta|$. $\qquad\square$

If a linear dynamical system generates a nearly periodic sequence, we can show the linear system has a specific structure as follows:

**Proposition 4.2.** *Let $M : \mathbb{R}^d \to \mathbb{R}^d$ be a linear map. Let $\mathbb{C}^d = \oplus_\alpha V_\alpha$ be the decomposition via generalized eigenspaces of $M$, where $\alpha$ runs over the eigenvalues of $M$ and $V_\alpha := \mathcal{N}((\alpha I - M)^d)$. Assume that there exists $\mu \geq 0$, for any $\theta \in \mathbb{R}^d$, $(M^t \theta)_{t=t_0}^{\infty}$ is $\mu$-nearly periodic for some $t_0 \in \mathbb{N}$. Let $\theta = \sum_\alpha \theta_\alpha \in \oplus_\alpha V_\alpha$. Then, each eigenvalue $\alpha$ such that $\theta_\alpha \neq 0$ satisfies $|\alpha| \leq 1$, in addition, if $|\alpha| = 1$ and $\theta_\alpha \neq 0$, $M\theta_\alpha = \alpha\theta_\alpha$.*

*Proof.* We note that $\{M^t v\}_{t\geq 0}$ is bounded for any $v \in \mathbb{R}^d$ by the assumption on $M$. Thus, $M$ cannot have an eigenvalue of magnitude greater than 1. We show that $\alpha$ is in the form of $\alpha = e^{i2\pi q}$ for some $q \in \mathbb{Q}$ if $|\alpha| = 1$. Suppose that $\alpha = e^{i2\pi\gamma}$ for an irrational number $\gamma \in \mathbb{R}$. Then, for an eigenvector $w$ for $\theta_\alpha$ with $\|w\|_{\mathbb{R}^d} > \mu$, $\{M^t w\}_{t>t_0}$ cannot become a $\mu$-periodic sequence. Thus, we conclude $\alpha = e^{i2\pi q}$ for some $q \in \mathbb{Q}$. Next, we show $M\theta_\alpha = \alpha\theta_\alpha$ if $|\alpha| = 1$ and $\theta_\alpha \neq 0$. Suppose $(M - \alpha I)\theta_\alpha \neq 0$. Since $(M - \alpha I)^d \theta_\alpha = 0$, there exists $1 \leq d' < d$ such that $(M - \alpha I)^{d'+1}\theta_\alpha = 0$ but $(M - \alpha I)^{d'}\theta_\alpha \neq 0$. Let $w' := (M - \alpha I)^{d'-1}\theta_\alpha$. Then, we see that $(M - \alpha I)^2 w' = 0$ but $(M - \alpha I)w' \neq 0$. By direct computation, we see that

$$\|M^t w'\|_{\mathbb{R}^d} = \|(M - \alpha I + \alpha I)^t w'\|_{\mathbb{R}^d} \geq t\|(M - \alpha I)w'\|_{\mathbb{R}^d} - \|w'\|_{\mathbb{R}^d}.$$

Thus, we have $\|M^t w'\|_{\mathbb{R}^d} \to \infty$ as $t \to \infty$, which contradicts the fact that $\{M^t w'\}_{t\geq 0}$ is a bounded sequence. The last statement is obvious. $\qquad\square$

Let $W$ be a linear subspace of $\mathbb{C}^d$ generated by the trajectory $\{M\theta, M^2\theta, \ldots\}$ and denote $\dim(W)$ by $d_0$. Note that restriction of $M$ to $W$ induces a linear map from $W$ to $W$. We denote by $M_\theta$ the induced linear

map from $W$ to $W$. Let $W = \oplus_{\alpha \in \Lambda} W_\alpha$ be the decomposition via the generalized eigenspaces of $M_\theta$, where $\Lambda$ is the set of eigenvalues of $M_\theta$ and $W_\alpha := \mathcal{N}((\alpha I - M_\theta)^{d_0})$. We define

$$W_{=1} := \oplus_{|\alpha|=1} W_\alpha,$$
$$W_{<1} := \oplus_{|\alpha|<1} W_\alpha.$$

Then, we have the following statement as a corollary of Proposition 4.2:

**Corollary 4.3.** *There exist linear maps $M_1, M_{<1} : W \to W$ such that*

1. *$M_\theta = M_1 + M_{<1}$,*

2. *$M_1 M_{<1} = M_{<1} M_1 = O$,*

3. *$M_1$ is diagonalizable and any eigenvalue of $M_1$ is of magnitude 1, and*

4. *any eigenvalue of $M_{<1}$ is of magnitude smaller than 1.*

*Proof.* Let $p : W \to W_{=1}$ be the projection and let $i : W_{=1} \to W$ be the inclusion map. We define $M_1 := i M_\theta p$. We can construct $M_{<1}$ in the similar manner and these matrices are desired ones. $\square$

**Example 4.2.** Let $G$ be a finite group and let $\rho$ be a finite dimensional representation of $G$, namely, a group homomorphism $\rho : G \to \mathrm{GL}_m(\mathbb{C})$, where $\mathrm{GL}_m(\mathbb{C})$ is the set of complex regular matrices of size $m$. Fix $g \in G$. Let $B \in \mathbb{C}^{n \times n}$ be a matrix whose eigenvalues have magnitudes smaller than 1. We define a matrix of size $m + n$ by

$$M := \begin{pmatrix} \rho(g) & 0 \\ 0 & B \end{pmatrix}.$$

Then, $(M^t x)_{t=t_0}^\infty$ is a $\mu$-nearly periodic sequence for any $\mu > 0$, $x \in \mathbb{C}^m$, and sufficiently large $t_0$. Moreover, we know that the length of the nearly period is $|G|$. We treat the permutation of variables in $\mathbb{R}^d$ in the simulation experiment (see Section 6), namely the case where $G$ is the symmetric group $\mathfrak{S}_d$ and $\rho$ is a homomorphism from $G$ to $\mathrm{GL}_d(\mathbb{C})$ defined by $\rho(g)((x_j)_{j=1}^d) := (x_{g(j)})_{j=1}^d$, which is the permutation of variable via $g$.

## 4.2 Problem setting

Here, we state our problem settings. We use the notation introduced in the previous sections. First, we summarize our technical assumptions as follows:

**Assumption 1** (Conditions on arms). *The set of arms $D$ contains the unit hypersphere.*

**Assumption 2** (Assumptions on noise). *The noise sequence $\{\eta_t\}_{t=1}^\infty$ is conditionally $R$-sub-Gaussian ($R \in \mathbb{R}_{\geq 0}$), i.e., given $t$,*

$$\forall \lambda \in \mathbb{R}, \ \mathbb{E}\left[e^{\lambda \eta_t} | \mathcal{F}_{t-1}\right] \leq e^{\frac{\lambda^2 R^2}{2}},$$

*and $\mathbb{E}[\eta_t | \mathcal{F}_{t-1}] = 0$, $\mathrm{Var}[\eta_t | \mathcal{F}_{t-1}] \leq R^2$, where $\{\mathcal{F}_\tau\}_{\tau \in \mathbb{N}}$ is an ascending family and we assume that $x_1, \ldots, x_{\tau+1}, \eta_1, \ldots, \eta_\tau$ are measurable with respect to $\mathcal{F}_\tau$.*

**Assumption 3** (Assumptions on dynamical systems). *There exists $\mu > 0$ such that for any $\theta \in \Theta$, the sequence $(f^t(\theta))_{t=t_0}^\infty$ is $\mu$-nearly periodic of length $L$ for some $t_0 \in \mathbb{N}$. We denote by $B_\theta$ the radius of the smallest ball containing $\{f^t(\theta)\}_{t=0}^\infty$.*

**Remark 4.4.** Assumption 1 excludes the lower bound arguments of the minimally required samples for our work since taking sufficiently large vector (arm) makes the noise effect negligible. Considering more restrictive conditions for discussing lower bounds is out of scope of this work.

Then, our questions are described as follows:

- Can we estimate the length $L$ from the collection of rewards $(r_t(x_t))_{t=1}^T$ efficiently ?

---

**Algorithm 1** Period estimation (DFT)

---

    **Input**: Current time $t_0 \in \mathbb{Z}_{>0}$; $T_p$; $\epsilon > 0$; $L_{\max} > 1$; orthogonal basis $\{\mathbf{u}_1, \ldots, \mathbf{u}_d\}$ of $\mathbb{R}^d$
    **Output**: Estimated length $\hat{L}$
1: $\ell \leftarrow 1$; $\beta \leftarrow 1$
2: **for** $m = 1, \ldots, d$ **do**
3:     **for** $t = t_0, t_0 + 1, \ldots, t_0 + T_p - 1$ **do**
4:         Sample arm $\mathbf{u}_m$ and observe $r_t(\mathbf{u}_m)$
5:     **end for**
6:     **while** $\ell \cdot \beta \leq L_{\max}$ **do**
7:         $\ell \leftarrow \ell + 1$
8:         **for** $(s, b) = (0, 1), (0, 2), \ldots, (0, \ell - 1), (1, 1), (1, 2), \ldots, (\beta - 1, \ell - 1)$ **do**
9:             **if** $\left| \mathcal{R}\left( (r_{t_0 + s + \beta t}(\mathbf{u}_m))_{t=1}^{\lfloor T_p/\beta \rfloor}; b/\ell \right) \} \right|^2 > \epsilon$ **then**
10:                $\beta \leftarrow \beta \ell$
11:                $\ell \leftarrow 1$
12:                Break
13:             **end if**
14:         **end for**
15:     **end while**
16:     $t_0 \leftarrow t_0 + T_p$
17: **end for**
18: $\hat{L} \leftarrow \beta$

---

- If we assume the dynamical system is linear, can we further obtain the eigenvalues of $f$ from a collection of rewards ?

- How many samples do we need to provably estimate the length $L$ or eigenvalues of $f$ ?

We will answer these questions in the following sections and via simulation experiments.

## 5 Algorithms and theory

With the above settings in place, we present a (computationally efficient) algorithm for each presented problem, and show its sample complexity for estimating certain information.

### 5.1 Period estimation

Here, we describe an algorithm for period estimation followed by its theoretical analysis. The overall procedure is summarized in Algorithm 1. Line 8 executes *concurrent application of filterings*. To analyze the sample complexity of this estimation algorithm, we first introduce an exponential sum that plays a key role:

**Definition 5.1.** For a positive rational number $q \in \mathbb{Q}_{>0}$ and $T$ complex numbers $a_1, \ldots, a_T \in \mathbb{C}$, we define

$$\mathcal{R}\left( (a_t)_{t=1}^T; q \right) := \frac{1}{T} \sum_{j=1}^{T} a_j e^{i2\pi q j}.$$

For a $\mu$-nearly periodic sequence $\mathbf{a} := (a_t)_{t=1}^{\infty}$ of length $L$, we define the supremum of the standard deviations of the $L$ sequential data of $\mathbf{a}$:

$$\sigma_L(\mathbf{a}) := \sup_{t_0 \geq 1} \sqrt{\frac{1}{L} \sum_{t=t_0}^{t_0 + L - 1} \left| a_t - \frac{1}{L} \sum_{j=t_0}^{t_0 + L - 1} a_j \right|^2}.$$

The exponential sum $\mathcal{R}(\cdot;\cdot)$ can extract a divisor of the nearly period of a $\mu$-nearly periodic sequence if $\mu$ is "sufficiently smaller" than the variance of the sequence even when the sequence is contaminated by noise; more precisely, we have the following lemma:

**Lemma 5.1.** *Let* $\mathbf{a} := (a_j)_{j=1}^\infty$ *be a* $\mu$-*nearly periodic sequence of length* $L$. *Then, we have the following statements:*

1. *if* $L > 1$, *then there exists* $s \in \mathbb{Z}_{>0}$ *with* $s < L$ *such that*

$$\left| \mathcal{R}\left( (a_j)_{j=1}^T ; s/L \right) \right| > \sqrt{\frac{\sigma^2 - 2\mu\sigma}{L}} - \mu - \frac{L \sup_{t \geq 1} |a_t|}{T}, \tag{5.1}$$

2. *if* $\beta$ *is not a divisor of* $L$, *then for any* $\alpha \in \mathbb{Z}_{>0}$,

$$\left| \mathcal{R}\left( (a_j)_{j=1}^T ; \alpha/\beta \right) \right| < \mu + \frac{L^2 \mathcal{C}_0 (\mu + \sup_{t \geq 1} |a_t|)}{T}, \tag{5.2}$$

*where* $\mathcal{C}_0 := 1 + 2/\sqrt{2}\pi(3/4)^{\pi^2/6} = 1.72257196806914....$

*Proof.* As $(a_t)_{t=1}^\infty$ is $\mu$-almost periodic, there exist $(b_t)_{t=1}^\infty$ of period $L$ and $(c_t)_{t=1}^\infty$ with $\sup_{t \geq 1} |c_t| < \mu$ such that $a_t = b_t + c_t$.

First, we prove (5.1). Let

$$\tilde{b} := \frac{1}{L} \sum_{t=1}^L b_t, \quad \hat{b}_q := \frac{1}{L} \sum_{t=1}^L b_t e^{i2\pi t q}, \quad (q \in \mathbb{Q}).$$

We claim that

$$L \cdot \sup\{|\hat{b}_{s/L}|^2\}_{s=1}^{L-1} > \sum_{s=1}^{L-1} |\hat{b}_{s/L}|^2 = \frac{1}{L} \sum_{t=1}^L |b_t - \tilde{b}|^2 \geq \sigma^2 - 2\mu\sigma. \tag{5.3}$$

In fact, the first inequality is obvious. The equality follows from the Plancherel formula for a finite abelian group (see, for example, (Serre, 1977, Excercise 6.2)). As for the last inequality, take arbitrary $t_0 \in \mathbb{Z}_{>0}$ and define $\tilde{a} = L^{-1} \sum_{t=t_0}^{t_0+L-1} a_t$ and $\tilde{c} = L^{-1} \sum_{t=t_0}^{t_0+L-1} c_t$. Then, we have

$$\frac{1}{L} \sum_{t=1}^L |b_t - \tilde{b}|^2 \geq \frac{1}{L} \sum_{t=t_0}^{t_0+L-1} |a_t - \tilde{a}|^2 + \frac{1}{L} \sum_{t=t_0}^{t_0+L-1} |c_t - \tilde{c}|^2 - \frac{2}{L} \sum_{t=t_0}^{t_0+L-1} |a_t - \tilde{a}| \cdot |c_t - \tilde{c}|$$

$$\geq \frac{1}{L} \sum_{t=t_0}^{t_0+L-1} |a_t - \tilde{a}|^2 + \frac{1}{L} \sum_{t=t_0}^{t_0+L-1} |c_t - \tilde{c}|^2$$

$$- 2\sqrt{\frac{1}{L} \sum_{t=t_0}^{t_0+L-1} |a_t - \tilde{a}|^2} \cdot \sqrt{\frac{1}{L} \sum_{t=t_0}^{t_0+L-1} |c_t - \tilde{c}|^2}$$

$$\geq \frac{1}{L} \sum_{t=t_0}^{t_0+L-1} |a_t - \tilde{a}|^2 - 2\mu\sigma.$$

Here, we used the Cauchy-Schwartz inequality in the second inequality. Since $t_0$ is arbitrary, we have (5.3). Let $s \in \text{argmax}_{s=1,\ldots,L-1} |\hat{b}_{s/L}|^2$ and let $q := s/L$. Let $T = LT' + \gamma$ for some $T', \gamma \in \mathbb{N}$ with $0 \leq \gamma < L$. Then, we have

$$\left| \mathcal{R}\left( (a_j)_{j=1}^T ; q \right) \right| \geq \left| \frac{1}{T'} \sum_{t=0}^{T'-1} \left[ \frac{1}{L} \sum_{s=1}^L a_{Lt+s} e^{i2\pi qs} \right] \right| - \frac{L \sup_{t \geq 1} |a_t|}{T}$$

$$> |\hat{b}_q| - \mu - \frac{L \sup_{t \geq 1} |a_t|}{T}$$

$$\geq \sqrt{\frac{\sigma^2 - 2\mu\sigma}{L}} - \mu - \frac{L \sup_{t \geq 1} |a_t|}{T}.$$

Next, we prove (5.2). As in the same computation as above, we have

$$\left|\mathcal{R}\left((a_j)_{j=1}^T; \alpha/\beta\right)\right| \leq \frac{LT'}{T}\left|\frac{1}{T'}\sum_{t=0}^{T'-1}e^{i2\pi\alpha Lt/\beta}\left[\frac{1}{L}\sum_{s=1}^{L}a_{Lt+s}e^{i2\pi\alpha s/\beta}\right]\right| + \frac{L\sup_{t>0}|a_t|}{T}$$

$$< \frac{LT'}{T}\left|\frac{1}{T'}\sum_{t=0}^{T'-1}e^{i2\pi\alpha Lt/\beta}\right| \cdot \left|\frac{1}{L}\sum_{s=1}^{L}b_s e^{i2\pi\alpha s/\beta}\right| + \mu + \frac{L\sup_{t>0}|a_t|}{T}$$

$$< \frac{2}{\left|1-e^{i2\pi\alpha L/\beta}\right|}\cdot\left(\mu+\sup_{t\geq 1}|a_t|\right)\cdot\frac{L}{T} + \mu + \frac{L\sup_{t>0}|a_t|}{T}.$$

Since $|1 - e^{i2\pi a}| \geq \sqrt{2}(1-a^2)^{\pi^2/6}a$ for $a \in (0,1)$ by (Chesneau & Bagul, 2020, Proposition 3.2), we have

$$\left|\mathcal{R}\left((a_j)_{j=1}^T; \alpha/\beta\right)\right| < \left(\mu+\sup_{t\geq 1}|a_t|\right)\frac{2\beta L}{\sqrt{2}(3/4)^{\pi^2/6}T} + \mu + \frac{L\sup_{t\geq 1}|a_t|}{T}$$

$$< \mu + \frac{L\beta\mathcal{C}_0(\mu+\sup_{t\geq 1}|a_t|)}{T}.$$

$\square$

Then, we obtain the explicit lower bound of the samples for period estimation:

**Proposition 5.2.** *Let $\mathbf{a} := (a_t)_{t=1}^{\infty}$ be a $\mu$-nearly periodic sequence of length $L$. Fix a positive integer $L_{\max} > 1$ with $L \leq L_{\max}$, $\delta \in (0,1)$, $\xi \in (0,1)$, and $\sigma_0 > 0$. Let $(\eta_t)_{t=1}^{\infty}$ be a noise sequence satisfying Assumption 2. Put $\gamma := 1/(1 + \sqrt{4L_{\max}+1})$ and $\lambda := \mu/(\sigma_0\gamma)$. We define*

$$\varepsilon := \sigma_0\gamma\xi.$$

*If $\mu/(\gamma\xi) < \sigma_0 \leq \sigma_L(\mathbf{a})$, then, for any*

$$T \geq \frac{8L_{\max}R^2\log(4/\delta)}{\sigma_0^2(\xi-\lambda)^2} + \frac{36L_{\max}^{5/2}\sup_{t\geq 1}|a_t|}{\sigma_0(\xi-\lambda)},$$

*the set of rational numbers*

$$S_{T,\varepsilon} := \left\{q \in \mathbb{Q}\cap(0,1): qL \in \mathbb{Z}_{>0} \text{ and } \left|\mathcal{R}\left((a_t+\eta_t)_{t=1}^T; q\right)\right| > \varepsilon\right\}$$

*is non-empty with probability at least $1 - \delta$.*

If we apply several collections of rewards $(r_t(x_t))_{t=1}^T$ for sufficiently large $T$ indicated in Proposition 5.2, we obtain various divisors of $L$. Finally, we provide the precise inputs and output of Algorithm 1 in the following Theorem:

**Theorem 5.3.** *Suppose Assumptions 1, 2 and 3 hold. Let $r \in [0,1)$ be a non-negative real number, and suppose $\rho > 0$ and $\delta \in (0,1)$ are given. Fix a positive integer $L_{\max} > 1$ with $L \leq L_{\max}$. We define*

$$\varepsilon := \frac{\rho}{6\sqrt{d}L_{\max}}. \tag{5.4}$$

*Assume that $r\varepsilon \geq \mu$. Let $T_p$ be an integer satisfying*

$$T_p \geq \frac{72dAL_{\max}^2}{\rho^2(1-r)^2} + \frac{108B_\theta\sqrt{d}L_{\max}^3}{\rho(1-r)}, \tag{5.5}$$

*where $A := R^2\log(4dL_{\max}^2\log L_{\max}/\delta)$. Then, the output $\hat{L}$ of Algorithm 1 is a $(\rho, \sqrt{d})$-anp of $(f^t(\theta))_{t=t_0}^{\infty}$ with probability at least $1 - \delta$.*

If $\mu$ is sufficiently small, we may set $r$ as a small positive number, in particular $r = 0$ if the system is periodic.

**Remark 5.4.** If random arm selection is adopted rather than the orthogonal basis, it may underestimate an error margin on some dimensions, which could lead to the nearly period with much larger error margin than expected; considering failure probability of such a case may potentially produce a variant of our algorithm.

---

**Algorithm 2** Eigenvalue estimation

---

**Input**: Effective sample size $N \in \mathbb{Z}_{>0}$; threshold $\gamma(N)$
**Output**: Matrix $A_1(N)\hat{A}_0(N)^\dagger$

1: Independently draw random unit vectors $\tilde{x}_m, \quad m \in [d]$, from uniform distribution over the unit sphere in $\mathbb{R}^d$.
2: Wait $N$ time steps.
3: **for** $t = N + 1, \cdots, N + 2Nd^2$ **do**
4: $\quad m_0 \leftarrow \{(t - N - 1) \mod 2d^2\} + 1$
5: $\quad m \leftarrow \lceil m_0/2d \rceil$
6: $\quad$ Sample arm $x_t = \tilde{x}_m$ and observe $\tilde{r}_t := r_t(\tilde{x}_m)$.
7: **end for**
8: Construct matrices $A_0(N)$ and $A_1(N)$ as in (5.6), respectively.
9: Obtain the low rank approximation $\hat{A}_0(N)$ of $A_0(N)$ via SVD with the threshold $\gamma(N)$.
10: Output $A_1(N)\hat{A}_0(N)^\dagger$.

---

## 5.2 Eigenvalue estimation

If the underlying system has certain structures, more detailed information about the system is expected to be obtained. In this section, we assume the following condition, linearity of the underlying dynamical system $f$ on $\Theta$, in addition to Assumption 1, 2, and 3:

**Assumption 4** (Linear dynamical systems). *The dynamical system $f : \Theta \to \Theta$ is linear and is represented by a matrix $M \in \mathbb{R}^{d \times d}$.*

Let $\mathbb{C}^d = \oplus_\alpha V_\alpha$ be the decomposition via generalized eigenspaces of $M$, where $\alpha$ runs over the eigenvalues of $M$ and $V_\alpha := \mathcal{N}((\alpha I - M)^d)$. We describe $\theta = \sum_\alpha \theta_\alpha$ with $\theta_\alpha \in V_\alpha$. We remark that an eigenvalue $\alpha$ of $M$ such that $\theta_\alpha \neq 0$ is in the form of $e^{2\pi i \ell/L}$ unless $|\alpha| < 1$ by Proposition 4.2.

Our objective is to estimate some of, if not all of, the eigenvalues of $M$ with high probability within some error that decreases by the sample size. To this end, we define the meaningful subset of eigenvalues of $M$.

**Definition 5.2** $((\theta, k)$-distinct eigenvalues). For a vector $\theta \in \mathbb{C}^d$ and $k \in \mathbb{Z}_{>0}$, we define a $(\theta, k)$-distinct eigenvalue by an eigenvalue $\beta$ of $M^k$ such that $|\beta| = 1$ and $\theta_\beta \neq 0$.

In our case, starting from a vector $\theta$, the effect of the eigenvalues that are not of $(\theta, d)$-distinct eigenvalues of $M$ may not be observable. Basically, once being able to ignore the effects of eigenvalues of magnitudes less than 1, the system becomes nearly periodic and we aim at estimating $(\theta, d)$-distinct eigenvalues as we obtain more samples.

Our eigenvalue estimation algorithm is summarized in Algorithm 2; it maintains the following matrices. For $N \in \mathbb{Z}_{>0}$, $d$ random unit vectors $\tilde{x}_1, \ldots, \tilde{x}_d$, and $s = 0, 1$, we define the matrix $A_s(N) \in \mathbb{C}^{d \times d}$ so that its $(k, \ell)$ element is given by

$$\sum_{j=0}^{N-1} r_{2(k-1)d+sd+2d^2j+N+\ell}(\tilde{x}_k) e^{\frac{i2\pi j^2}{4L}}. \tag{5.6}$$

That is, after $N$ steps, the reward multiplied by $\exp(i2\pi j^2/4L)$ is placed from the top row of $A_0$ and then the top row of $A_1$, followed by the second rows of them, and so on. Then, those values are summed up for every $2d^2$ steps or every $j$th cycle. Here, after throwing away $N$ samples, the effects of eigenvalues of magnitude less than 1 become negligible, and the trajectory becomes nearly periodically behaved under Assumption 4. The rest of the samples is used to average out the observation noise while maintaining some meaningful information about $M$.

The aforementioned exponential sum can be characterized by the Weyl-type sum of matrices, a key machinery for our algorithm, which we define below:

**Definition 5.3.** Let $W$ be a linear space and let $M_1, \ldots, M_N$ for $N \in \mathbb{Z}_{>0}$ be linear maps on $W$. For $L \in \mathbb{Z}_{>0}$, we define

$$\mathscr{W}((M_1, \ldots, M_N)) := \frac{1}{N} \sum_{j=0}^{N-1} M_{j+1} e^{\frac{i2\pi j^2}{4L}}.$$

**Remark 5.5.** Let $E_{s,n} := (\eta_{2di+N+sd+j+1+2d^2 n})_{i,j=0,\ldots,d-1}$ be a noise matrix for $s = 0, 1$. Let

$$X := \begin{pmatrix} \tilde{x}_1^\top M^1 \\ x_2^\top M^{2d+1} \\ \vdots \\ \tilde{x}_d^\top M^{2(d-1)d+1} \end{pmatrix} \text{ and } K := (M\theta, \ldots, M^d\theta).$$

Then, $A_s(N)$ has an alternative description as follows:

$$A_s(N) = X\mathscr{W}\left((M^{2d^2 j})_{j=0}^{N-1}\right) M^{sd+N-1}K + \mathscr{W}\left((E_{s,j})_{j=0}^{N-1}\right). \tag{5.7}$$

As in Proposition 5.7 below, the Weyl-type sum has a crucial property. Define $\kappa \geq 1$ by

$$\kappa := \inf_P \left\{ \|P\| \|P^{-1}\| : P^{-1}MP = J_M \right\},$$

where $J_M$ is a Jordan normal form of $M$. Also, we define $\Delta \in (0, 1]$ to be a value such that, for any eigenvalue $\alpha$ of $M$ satisfying $|\alpha| < 1$, $|\alpha| \leq 1 - \Delta$ (define $\Delta = 1$ if no such eigenvalue exists). Note by Proposition 4.2, the existence of such spectral gap is guaranteed without any further assumptions.

Roughly speaking, Algorithm 2 estimates "$A_1(N)A_0(N)^{-1} = XM^dX^{-1}$". Of course, the formula in "..." is not valid as $X$, $K$, and the Weyl-type sum are not necessarily invertible and we cannot recover full information of $M^d$ in general. However, we can still reconstruct information of $M^d$ restricted on the eigenspaces for $(\theta, d)$-distinct eigenvalues.

To see this, we introduce the Weyl sum and its lower bound:

**Lemma 5.6** (Lower bound on the Weyl sum Bourgain (1993); Oh). *Define the Weyl sum by*

$$\mathscr{W}(N, b, q) := \sum_{j=0}^{N} e^{i2\pi\left(j\frac{2b}{4q} + j^2 \frac{1}{4q}\right)},$$

*for some $b \in \mathbb{N}$, $q \in \mathbb{Z}_{>0}$, $b < q$ and $N \in \mathbb{Z}_{>0}$. Then, for $N \geq 16q^2$, it holds that*

$$|\mathscr{W}(N, b, q)| \in \Omega\left(\frac{N}{\sqrt{q}}\right).$$

*Proof.* It is immediate from Proposition 3.1 in Oh because $\gcd(1, 4q) = 1$, $4q \equiv 0 \mod 4$, and $2b$ is even. $\quad\square$

We define $C_{\text{ws}}(L) > 0$ by

$$C_{\text{ws}}(L) := \inf\left\{C > 0 : C^{-1} < \left|\frac{1}{N}\mathscr{W}(N, b, L)\right| < C \text{ for any } N \geq 16L^2, 0 \leq b < L\right\}.$$

Then, we have the following proposition:

**Proposition 5.7.** *Let $M_1$ and $M_{<1}$ be matrices as in Corollary 4.3. We define linear maps on $W$ by*

$$Q_N(M_1) := \mathscr{W}\left((M_1^{2d^2 j})_{j=0}^{N-1}\right)$$

$$Q_N(M_{<1}) := \mathscr{W}\left((M_{<1}^{2d^2 j})_{j=0}^{N-1}\right).$$

*Then, we have the following statements:*

1. $\mathscr{W}\left((M_\theta^{2d^2j})_{j=0}^{N-1}\right) = Q_N(M_1) + Q_N(M_{<1})$,

2. for any $N \geq 16L^2$, $Q_N(M_1)$ is invertible on $W_{=1}$,

3. for any $r \geq 0$ and for any $N \geq 16L^2$, $\|M_1^r Q_N(M_1)|_{W_{=1}}\|, \|(M_1^r Q_N(M_1))|_{W_{=1}}^{-1}\| \leq \kappa C_{\mathrm{ws}}(L)$,

4. for any $r \geq 2(d-1)$, we have

$$\|M_{<1}^r Q_N(M_{<1})\| \leq \frac{d^2 \kappa e^{-\Delta(r-d+1)}}{N\Delta^{d-1}}.$$

*Proof.* We prove 1. By the properties 1 and 2 in Corollary 4.3, we have $\mathscr{W}((M^{2d^2j})_{j=0}^{N-1}) = Q_N(M_1) + Q_N(M_{<1})$. Next, we prove 2. When we regard $Q_N(M_1)|_{W_{=1}}$ as a linear map on $W_{=1}$, it is represented as a diagonal matrix $\mathrm{diag}(\mathscr{W}(N, b_1, L), \ldots, \mathscr{W}(N, b_m, L))$, where $m = \dim W_{=1}$. Therefore, by Lemma 5.6, $Q_N(M_1)$ is a bijective linear map on $W_{=1}$. Next, we prove 3. We estimate $\|M_1^r Q_N(M_1)\|$. Let $p : \mathbb{C}^d \to W$ be the orthogonal projection and let $i : W \to \mathbb{C}^d$ be the inclusion map. Let $\tilde{M}_1 := iM_1 p \in \mathbb{C}^{d \times d}$. Then, we have

$$\|Q_N(M_1)\| = \|Q_N(\tilde{M}_1)\| \leq \kappa C_{\mathrm{ws}}(L).$$

Next, we estimate $\|M_{<1}^r Q_N(M_{<1})\|$. Let $\tilde{M}_{<1} := iM_{<1}p$. Then, $iM_{<1}^r Q_N(M_{<1})p = \mathscr{W}((\tilde{M}_{<1}^{2d^2j+r})_{j=0}^{N-1})$ and $\|iM_{<1}^r Q_N(M_{<1})p\| = \|M_{<1}^r Q_N(M_{<1})\|$. Let $P$ be a regular matrix such that

$$\tilde{M}_{<1} = PJP^{-1},$$

where $J$ is the Jordan normal form. Then, for $r \geq 2(d-1)$, we see that

$$\begin{aligned}
\|M_{<1}^r Q_N(M_{<1})\| &= \|iM_{<1}^r Q_N(M_{<1})p\| \\
&\leq \frac{\kappa}{N} \sum_{j=0}^{N-1} \|J^{2d^2j+r}\| < \frac{d^2\kappa}{N} \sum_{j=0}^{\infty} \binom{2d^2j+r}{d-1}(1-\Delta)^{2d^2j+r-d+1} \\
&\leq \frac{d^2\kappa(1-\Delta)^{r-d+1}}{N\Delta^{d-1}} \leq \frac{d^2\kappa e^{-\Delta(r-d+1)}}{N\Delta^{d-1}}.
\end{aligned}$$

The second last inequality is proved as follows: for $|a| < 1$, $n \geq 1$ and $r \geq m \geq 0$,

$$\begin{aligned}
\left| \sum_{j=0}^{\infty} \binom{nj+r}{m} a^{nj-m+r} \right| &= \left| \frac{1}{m!} \frac{d^m}{dx^m} \sum_{j=0}^{\infty} x^{nj+r} \right|_{x=a} \\
&\leq \frac{1}{m!} \sum_{j=0}^{m} \binom{m}{j} \frac{r!}{(r-m+j)!} |a|^{r-m+j} \left| \frac{d^j}{dx^j} \frac{1}{1-x^n} \right|_{x=a} \\
&\leq \frac{1}{m!} \sum_{j=0}^{m} \binom{m}{j} \frac{r!}{(r-m+j)!} \frac{j!|a|^{r-m+j}}{(1-|a|)^j} = \sum_{j=0}^{m} \binom{r}{j} \frac{|a|^{r-m+j}}{(1-|a|)^j},
\end{aligned}$$

where, the last inequality follows from

$$\left| \frac{d^m}{dx^m} \frac{1}{1-x^n} \right| = \left| \sum_{\zeta^n=1} \frac{m!\zeta^{1-n}}{n(\zeta-x)^m} \right| \leq \frac{m!}{(1-|x|)^m}.$$

Thus, we have

$$\sum_{j=0}^{m} \binom{r}{j} \frac{|a|^{r-m+j}}{(1-|a|)^j} \leq |a|^{r-m} \sum_{j=0}^{r} \binom{r}{j} \left( \frac{|a|}{1-|a|} \right)^j \leq \frac{|a|^{r-m}}{(1-|a|)^m}.$$

$\square$

Proposition 5.7 plays an essential role in our analysis and guarantees that the information in $A_s(N)$ about $(\theta, d)$-distinct eigenvalues does not vanish while noise effects are canceled out. Now, we state our main theoretical result for the eigenvalue estimation algorithm. Before stating the theorem, we introduce lower rank approximation via the singular value threshold.

**Definition 5.4.** Let $A \in \mathbb{C}^{n \times m}$ be a matrix. Let $A = U \, (D \, O) \, V^*$ be a singular valued decomposition of $A$ where $U \in \mathbb{C}^{n \times n}$ and $V \in \mathbb{C}^{m \times m}$ are unitary matrices and $D := \mathrm{diag}(\sigma_1, \ldots, \sigma_r, \mathbf{o}^\top)$ is a diagonal matrix with nonnegative components. Let $\gamma > 0$. We define a low rank approximation $A_\gamma$ of $A$ via the singular value threshold $\gamma$ by the matrix $A_\gamma$ defined by $A_\gamma = U \, (D_\gamma \, O) \, V^*$, where, $D_\gamma := \mathrm{diag}(\mathbf{1}_{[\gamma, \infty)}(\sigma_1)\sigma_1, \ldots, \mathbf{1}_{[\gamma, \infty)}(\sigma_r)\sigma_r, \mathbf{o}^\top)$ and $\mathbf{1}_I$ is defined to be the characteristic function supported on $I \subset \mathbb{R}$.

Given this definition, we are ready to present the following main result:

**Theorem 5.8.** *Suppose Assumptions 1, 2, 3, and 4 hold. Given $\delta \in (0, 1]$, let the effective sample size*

$$N \geq \max \left\{ 16L^2, \frac{-(d-1)\log \Delta}{\Delta} + \frac{\log(B_\theta \kappa^2) + d + 6}{\Delta} + d \right\}, \tag{5.8}$$

*and $\gamma(N) = (\sqrt{4d^2 R^2 \log (4d^2/\delta)} + 1)/\sqrt{N}$. Then, there exists a matrix $A$ whose eigenvalues are zeros except for $(\theta, d)$-distinct eigenvalues of $M$, such that the output of Algorithm 2, i.e. $A_1(N)\hat{A}_0(N)^\dagger$, satisfies, with probability at least $1 - \delta$, that*

$$\left\| A - A_1(N)\hat{A}_0(N)^\dagger \right\| \leq C \left( \frac{R^2 (\log (1/\delta) + 1) + 1}{\sqrt{N}} \right). \tag{5.9}$$

*Here, $\hat{A}_0(N)^\dagger$ is the Moore-Penrose pseudo inverse of a lower rank approximation of $A_0(N)$ via the singular value threshold $\gamma(N)$. The constant $C > 0$ depends on $\theta$, $M$, $d$, $(\tilde{x}_m)_{m=1}^d$, and $C_{\mathrm{ws}}(L)$.*

We mention that by using the results shown in Song (2002), the bound on spectral norm (5.9) can be translated to the bounds on eigenvalues, where the constant depends on the form of $A$. As described in Theorem 5.8, the constant is not the absolute constant for any problem instance but depends on several factors; however, for the same execution, this rate is useful to judge how many samples one collects to estimate eigenvalues.

**Remark 5.9.** We note that we can reconstruct the $(\theta, 1)$-distinct eigenvalues of $M$ via Algorithm 2 using the following trick: Fix non-negative integer $r \geq 0$. Take $d + r$ random unit vectors $\tilde{x}_1, \ldots, \tilde{x}_{d+r}$. Then, for $s = 0, 1$, we may define a matrix $A_s(N; r) \in \mathbb{C}^{(d+r) \times (d+r)}$ so that its $(k, \ell)$ element is given by

$$\sum_{j=0}^{N-1} r_{2(k-1)(d+r)+s(d+r)+2(d+r)^2 j+N+\ell}(\tilde{x}_k) e^{\frac{i 2\pi j^2}{4L}}.$$

Then, we see that Algorithm 2 outputs a matrix $\tilde{A}(r)$ that well approximates the $(\theta, d+r)$-distinct eigenvalues of $M$ since the matrix $A_s(N; r)$ coincides with $A_s(N)$ in the case when we replace $M$ and $\theta$ with $M(r)$ and $\theta(r)$ defined by

$$M(r) := \begin{pmatrix} M & O \\ O & O \end{pmatrix} \in \mathbb{C}^{(d+r) \times (d+r)}, \theta(r) := \begin{pmatrix} \theta \\ \mathbf{o} \end{pmatrix} \in \mathbb{C}^{d+r}.$$

Let $r$ be an integer such that $r + d$ is prime to $L$ and fix a positive integer $m$ such that $m(r + d) \equiv 1 \mod L$. Then, the eigenvalues of $\tilde{A}(r)^m$ is close to those of $(\theta, 1)$-distinct eigenvalues if we take sufficiently large $N$.

## 6 Simulated experiments

In this section, we present simulated experiments that complement the theoretical claims. In particular, we conducted period estimations for an instance of LifeGame Conway et al. (1970), which is a special case

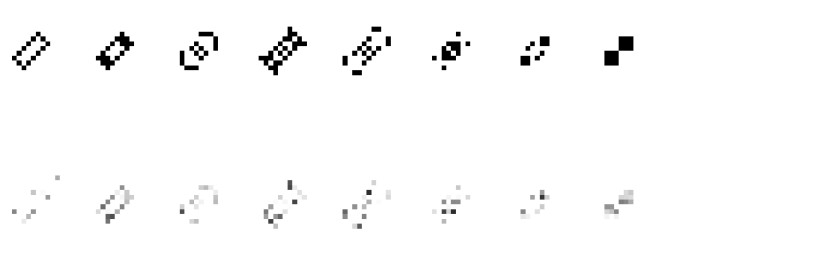 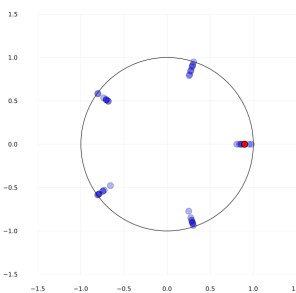

**Figure 5:** Left: Illustration of a period eight instance of LifeGame; (top) original transitions. (down) an instance of noisy observation. Right: $\mu$-nearly periodic dynamics (6.1).

of cellular automata Von Neumann et al. (1966), and for a nearly periodic toy system, and an eigenvalue estimation for a linear system, where some dimensions are for permutations and the rest is for shrinking.

**Period estimation: LifeGame.** We use a specific instance of LifeGame which is illustrated in Figure 5. As shown on the top eight pictures, starting from certain configuration of cells, it shows transitions of period eight. The sample size is computed as the smallest integer satisfying (5.5), and the threshold $\varepsilon$ is given by (5.4). To prevent the dimension from becoming too large, we used five cells that correctly display period eight; that is $d = 5$. Noise $\eta_t$ is given by i.i.d. Gaussian with proxy $R = 0.3$, and the down eight pictures of Figure 5 are some instances of noisy observations. We tested 50 different random seeds (i.e., 1, 51, 101, 151, ... , 2451), and computed the error rate (the number of runs producing a wrong estimate other than the fundamental period eight, which is divided by 50); and it was zero.

**Period estimation: Simple $\mu$-nearly periodic system.** We consider the following $\mu$-nearly periodic system that circulates over a circle with small variations:

$$r_{t+1} = \mu \left( \alpha \frac{r_t - 1}{\mu} - \lceil \alpha \frac{r_t - 1}{\mu} \rceil \right) + 1, \qquad \theta_{t+1} = \theta_t + \frac{2\pi}{L}, \tag{6.1}$$

where $r$ and $\theta$ are the radius and angle, and $\alpha \notin \mathbb{Q}$. We use $\mu = 0.001$, $L = 5$, and $\alpha = \pi$. Noise $\eta_t$ is drawn i.i.d. from the uniform distribution within $[-R, R]$ for $R = 0.3$. We tested 50 different random seeds (i.e., 1, 51, 101, 151, ... , 2451), and computed the error rate; and it was zero.

**Eigenvalue estimation: Permutation and shrink.** We use $d = 5$, and $M \in \mathbb{R}^{5 \times 5}$ is made such that 1) the first four dimensions are for permutation (i.e., each of row and column of $4 \times 4$ sub-matrix has only one nonzero element that is one.), and 2) the last dimension is simply shrinking; we gave 0.7 for $(5, 5)$-element of $M$. Initial vector $\theta_0$ and each arm $\tilde{x}_m$, $m \in [d]$, are uniformly sampled from the unit sphere in $\mathbb{R}^5$. The value $L$ is computed by $4! = 24$. We used the smallest integer $N$ that satisfies (5.8), multiplied by $C_{\text{sim}} > 0$. The results are shown in Table 1; it is observed that the more samples we use the more accurate the estimates become to $(\theta_0, 5)$-distinct eigenvalues of $M$. Noise $\eta_t$ is drawn i.i.d. from the uniform distribution within $[-R, R]$ for $R = 0.3$, and the Table 1 is of the random seed 1234. We also tested 50 different random seeds (i.e., 1, 51, 101, 151, ... , 2451) for $C_{\text{sim}} = 30$, and computed the mean absolute error between the true $(\theta_0, 5)$-distinct eigenvalues and their nearest estimated values; it was 0.0019, which is sufficiently small.

**Attempt on applications to more realistic data: Worm simulation.** Finally, we present an attempt on applications to more realistic situation, which will demonstrate some of the important aspects that need to be taken into account when naively applying our algorithms to the real world problems. In particular, we use OpenWorm (https://openworm.org/) (Szigeti et al., 2014) to simulate a worm motion (we ran more than around 10 hours for simulations); example images from this simulation is shown in Figure 6 (left). It is known that the behavior of worm has some periodicity (e.g., Ahamed et al. (2021)). We resized the images to $6 \times 6$ (i.e., $d = 36$), and downsampled the video by extracting one per eight images. Also, we repeat the same video in the way that the entire video still shows smooth flow (i.e., cut the video so that the end image is similar to the image in the beginning, and concatenate the same videos) so that we can sample a large number of data.

**Table 1:** Results for the eigenvalue estimation. The top-most row shows the true eigenvalues of $M^5$, and the second row shows its $(\theta_0, 5)$-distinct eigenvalues. From the third to sixth row, it shows the estimated eigenvalues for different values of $C_{\mathrm{sim}}$.

| eigenvalues of $M^5$ | 1.000 | 1.000 | $-0.500 - 0.866i$ | $-0.500 + 0.866i$ | 0.168 |
|---|---|---|---|---|---|
| $(\theta_0, 5)$ | 1.000 | 0 | $-0.500 - 0.866i$ | $-0.500 + 0.866i$ | 0 |
| when $C_{\mathrm{sim}} = 1$ | $1.000 + 0.008i$ | 0 | $-0.489 - 0.860i$ | $-0.510 + 0.869i$ | 0 |
| when $C_{\mathrm{sim}} = 5$ | $0.999 - 0.000i$ | 0 | $-0.495 - 0.867i$ | $-0.501 + 0.862i$ | 0 |
| when $C_{\mathrm{sim}} = 10$ | $1.000 + 0.003i$ | 0 | $-0.497 - 0.867i$ | $-0.501 + 0.864i$ | 0 |
| when $C_{\mathrm{sim}} = 30$ | $1.000 + 0.001i$ | 0 | $-0.500 - 0.866i$ | $-0.500 + 0.864i$ | 0 |

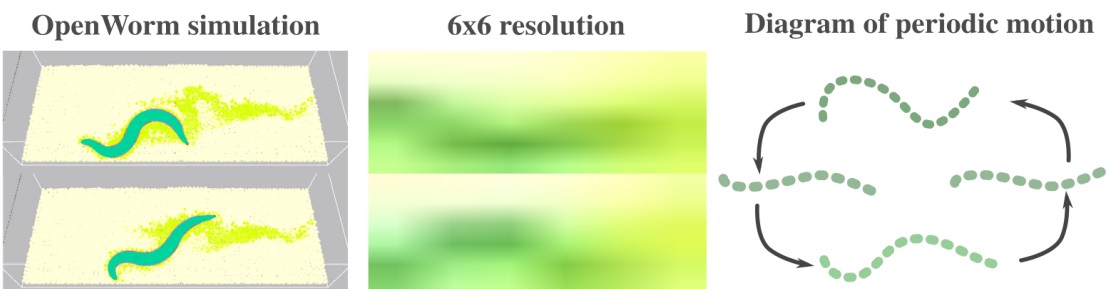

**OpenWorm simulation**    **6x6 resolution**    **Diagram of periodic motion**

**Figure 6:** Left: OpenWorm (https://openworm.org/) simulation of a worm motion. Middle: The extracted sequence of images with $6 \times 6$ resolution for our experiment. Right: Diagram of how the worm motion is viewed as a (nearly) periodic motion.

First, we assume we have no access to reasonable values for the hyperparameters, and show what happens when we use hyperparameters that do not reflect the target system. With such a choice of parameters (see Appendix F.4), the algorithm outputs an estimate that contradicts to the hyperparemters; with the value $L_{\max} = 25$, it outputs the estimate 40. In particular, there exist dimensions that output the estimates 8 and 20, from which the final estimate becomes 40. Note that, in this experiment, we use a fixed number of samples $T_p$ since we have no access to the reasonable hyperparameters. On the other hand, we increased the margin $\rho$ significantly, and then the output becomes $20 \leq L_{\max}$ where each dimension shows 1 or 20 in this case. In fact, one segment of our video data (recall we concatenate the same video to create a sequence of an indefinite number of images) consists of 120 images and contains 6 periods of sequence.

We mention that the aforementioned fact implies that the choice of hyperparameters may be practically validated by confirming that the outputs are consistent to the parameters.

Next, we assume that the system is linear and run the DMD over this canonical basis of the state space. Let $M_{\mathrm{DMD}}$ be the solution to the exact DMD over the video data, and we compute the eigenvalues of $M_{\mathrm{DMD}}^d$ and of the output of Algorithm 2. We tested the algorithm with the fixed hyperparameters given in Appendix F.4 and with varying number of $N$, namely, $10^3$, $10^4$, and $10^5$. For the cases of $10^3$ and $10^4$, we only obtained one eigenvalue which is close to $1 + 0i$ and the others were almost zero. For $N = 10^5$, we obtained three nonzero eigenvalues which are plotted in Figure 7 against the top four eigenvalues of $M_{\mathrm{DMD}}^d$. The actual nonzero eigenvalues and the estimated errors for all three cases are given in Table 2.

From the results, we argue that, for some real applications where the system is not exactly linear, the eigenvalue estimation algorithm may not work well although it does not output *insane* values.

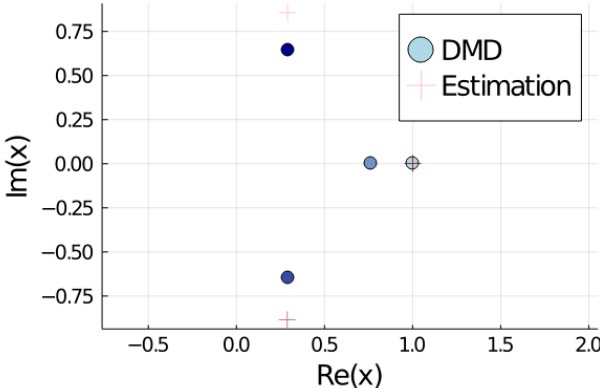

**Figure 7:** Estimated eigenvalues having large magnitudes (red) and the top four DMD eigenvalues (blue).

| $N$ | Nonzero eigenvalues | Errors |
|-----|---------------------|--------|
| $10^3$ | $1.000 + 0.002i$ | $0.199$ |
| $10^4$ | $1.000 + 0.002i$ | $0.199$ |
| $10^5$ | $1.001 + 0.001i$ $0.288 + 0.884i$ $0.290 + 0.867i$ | $0.172$ |

**Table 2:** Estimated nonzero eigenvalues for $N = 10^3, 10^4, 10^5$, and the total errors to the true DMD matrix $M_{\mathrm{DMD}}^d$.

## 7 Discussion and conclusion

This work proposed novel algorithms for estimating periodic information about the dynamical systems from bandit feedback contaminated by a sub-Gaussian noise. Since we consider a rather novel problem setup, the setting itself would be seen as a limitation of our work, and we present a potentially important list of future works below.

**Choice of hyperparameters such as $L_{\max}, N$:** If those hyperparameters are correctly chosen for nearly periodic systems, the outputs of our algorithms are consistent and reasonable across runs, which follows the theoretical insights; therefore, practically, one can check if the hyperparameters are properly chosen by running the algorithms. Refer to Section 6 for related discussions; in short, inappropriate choice of hyperparameters leads to contradicting outputs (e.g., period estimate larger than the assumed maximum value). On the other hand, studying if it is possible to theoretically ensure correctness of hyperparameters (e.g., by standard *doubling trick* (Cesa-Bianchi et al., 1997) for efficient guessing) or to identify non-periodic systems is an important future work.

**Extension to general spectrum information estimation problems:** We only impose nearly periodicity on the dynamical systems on top of the linearity for the eigenvalue estimation problem. Nevertheless, additional studies are needed to allow other forms of observations (not limited to bandit feedback) and non-periodic systems to consider eigenvalues with arguments of $2\pi$ times irrational numbers. Actually, the asymptotic bounds of the Weyl sum themselves are still valid when it is sufficiently well approximated by a rational number (Bourgain, 1993; Oh). Moreover, allowing control inputs would make eigenvalues of magnitudes smaller than one efficiently recoverable.

**Extensions to random dynamical systems:** This work studied deterministic dynamical systems; however, we conjecture that, under the condition that the variance of trajectories generated by a random dynamical system (RDS) (cf. Arnold (1998)) is sufficiently small, the similar estimation procedure is adopted for such RDSs. It is important to study if the estimation problem becomes easier when the system is driven by a particular noise as it could be treated as a random control input. Also, studying other dynamic structures such as the Lyapunov exponents for nonlinear systems is an interesting future work.

**Study of statistical estimation leveraged by other number theoretical results:** The proper use of exponential sums enables us to average out the noise while preserving particular information. Studying when this separation is feasible for different sets of information, noise, and class of problems should be important.

**Optimality of the results:** Although we gave a sufficient number of samples for provably guaranteeing (approximate) correctness of the estimates, it is unclear if our sample complexity is what one can best achieve under particular problem settings. Recall that the current problem settings without an assumption

on boundedness of the set of arms $D$ exclude the lower bound arguments of the minimally required samples as mentioned in Remark 4.4.

As such, instead of arguing the tightness, to justify the complexity of our algorithms, we hence mention that the very naive approach for (nearly) period estimation would cost the order of $L_{\max}!$ sample complexity. To see this, suppose we know the maximum possible (nearly) period $L_{\max}$ then it follows that $L_{\max}!$ is a true (nearly) period as a multiple of the fundamental (nearly) period. As such, one could employ the concentration of measures to average out the sample. With sufficiently many sequences of length $L_{\max}!$ depending on the rate of concentration, one could approximately reconstruct the original (noiseless) sequence of length $L_{\max}!$ from which one could estimate a desired $L \leq L_{\max}$ which is an (aliquot) nearly period.

Lower bound of the minimally required samples may be discussed by using similar arguments to the best arm identification problems (cf. Kaufmann et al. (2016)); however, our problems consider dynamical systems, and we have no clue on potential technical tools to deal with the arguments at this point.

## Acknowledgments

We thank the constructive comments by anonymous reviewers for improving this work. Motoya Ohnishi thanks Sham Kakade, Yoshinobu Kawahara, Emanuel Todorov, Jacob Sacks, Tomoharu Iwata, and Hiroya Nakao for valuable discussions and thanks Sham Kakade for computational supports. This work of Motoya Ohnishi, Isao Ishikawa, and Masahiro Ikeda was supported by JST CREST Grant, Number JPMJCR1913, including AIP challenge program, Japan. Also, Motoya Ohnishi is supported in part by Funai Overseas Fellowship. Isao Ishikawa is supported by JST ACT-X Grant, Number JPMJAX2004, Japan. This work of Yuko Kuroki was supported by Microsoft Research Asia and JST ACT-X JPMJAX200E.

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

# A    Proof of Proposition 5.2

*Proof.*

$$B := L_{\max} \sup_{t>0} |a_t|,$$
$$C := \sqrt{4R^2 \log(4/\delta)},$$
$$D := L_{\max}^2 \mathcal{C}_0 \left( \mu + \sup_{t\geq 1} |a_t| \right).$$

Then, $\sqrt{T} > 0$ satisfies the following inequality

$$\sqrt{T} \geq \max \left\{ \frac{C + \sqrt{C^2 + 4(\varepsilon - \mu)B}}{2(\varepsilon - \mu)}, \frac{C + \sqrt{C^2 + 4(\varepsilon - \mu)D}}{2(\varepsilon - \mu)} \right\}$$

if and only if

$$\varepsilon \leq 2\varepsilon - \mu - \frac{L_{\max} \sup_{t>0} |a_t|}{T} - \sqrt{\frac{4R^2 \log(4/\delta)}{T}}, \text{ and}$$
$$\varepsilon \geq \mu + \frac{L_{\max}^2 \mathcal{C}_0 (\mu + \sup_{t\geq 1} |a_t|)}{T} + \sqrt{\frac{4R^2 \log(4/\delta)}{T}}.$$

Since $L_{\max} \geq 2$, $D \geq B$, $\mu < \gamma \sup_{t\geq 1} |a_t|$, $\mathcal{C}_0(1+\gamma) < 3$, and

$$\varepsilon - \mu = \frac{\sigma_0(1-\lambda)}{\sqrt{L_{\max}}} \cdot \frac{\sqrt{L_{\max}}}{1 + \sqrt{4L_{\max} + 1}}$$
$$\geq \frac{\sigma_0(1-\lambda)}{2\sqrt{2}\sqrt{L_{\max}}},$$

$$\max \left\{ \frac{C + \sqrt{C^2 + 4(\varepsilon - \mu)B}}{2(\varepsilon - \mu)}, \frac{C + \sqrt{C^2 + 4(\varepsilon - \mu)D}}{2(\varepsilon - \mu)} \right\}$$
$$= \frac{C + \sqrt{C^2 + 4(\varepsilon - \mu)D}}{2(\varepsilon - \mu)}$$
$$\leq 2\sqrt{\frac{L_{\max}C^2}{4(\varepsilon - \mu)^2} + \frac{D}{(\varepsilon - \mu)}}$$
$$\leq 2\sqrt{\frac{2L_{\max}^2 C^2}{\sigma_0^2(\xi - \lambda)^2} + \frac{6\sqrt{2}L_{\max}^{5/2} \sup_{t\geq 1} |a_t|}{\sigma_0(\xi - \lambda)}}$$
$$\leq 2\sqrt{\frac{2L_{\max}^2 C^2}{\sigma_0^2(\xi - \lambda)^2} + \frac{9L_{\max}^{5/2} \sup_{t\geq 1} |a_t|}{\sigma_0(\xi - \lambda)}},$$

where we used $\sqrt{1 - 2\gamma} = 2\gamma\sqrt{L_{\max}}$. Since $2\varepsilon \leq \sqrt{(\sigma_0^2 - 2\mu\sigma_0)/L_{\max}}$, the statement follows from Lemma 5.1. $\square$

# B   Proof of Theorem 5.3

*Proof.* Let $\mathscr{K}$ be the set of all combinations $(m, \beta, s, q)$ such that $m \in [d]$, $\beta \in [L_{\max}]$, $s \in \{0, 1, \ldots, \beta - 1\}$ and $\{q = \alpha/\ell : \alpha \in \{1, \ldots, \ell - 1\}\}$. We remark that

$$
|\mathscr{K}| \leq \sum_{m=1}^{d} \sum_{\beta=1}^{L_{\max}} \sum_{\ell \leq L_{\max}/\beta} \sum_{\alpha=1}^{\ell-1} \sum_{s=0}^{\beta-1} 1 = d \sum_{\beta=1}^{L_{\max}} \sum_{\ell \leq L_{\max}/\beta} \beta(\ell - 1)
$$

$$
\leq d \sum_{\beta=1}^{L_{\max}} \frac{L_{\max}}{2} \left( \frac{L_{\max}}{\beta} - 1 \right)
$$

$$
\leq \frac{dL_{\max}^2 \log L_{\max}}{2}.
$$

Also, let $\mathcal{E}_\kappa^{T_p}$ be the event such that, for the combination $\kappa \in \mathscr{K}$,

$$
\left| \frac{1}{\lfloor T_p/\beta \rfloor} \sum_{j=0}^{\lfloor T_p/\beta \rfloor - 1} \left\{ \eta_{t_0 + \beta j + s} e^{\frac{i 2\pi \alpha j}{\ell}} \right\} \right| \leq \sqrt{\frac{4R^2 \log \left( 4dL_{\max}^2 \log L_{\max}/\delta \right)}{\lfloor T_p/\beta \rfloor}}.
$$

Because the error sequence satisfies conditionally $R$-sub-Gaussian, using Lemma D.1 and from the fact that any subsequence of the filtration $\{\mathcal{F}_\tau\}$ is again a filtration, we obtain, for each $\kappa \in \mathscr{K}$,

$$
\Pr\left[ \mathcal{E}_\kappa^{T_p} \right] \geq 1 - \frac{\delta}{dL_{\max}^2 \log L_{\max}}.
$$

Define $\mathcal{E}^{T_p} := \bigcap_{\kappa \in \mathscr{K}} \mathcal{E}_\kappa^{T_p}$. Then, it follows from the Fréchet inequality that

$$
\Pr\left[ \mathcal{E}^{T_p} \right] = \Pr\left[ \bigcap_{\kappa \in \mathscr{K}} \mathcal{E}_\kappa^{T_p} \right] \geq |\mathscr{K}| \left( 1 - \frac{\delta}{dL_{\max}^2 \log L_{\max}} \right) - (|\mathscr{K}| - 1)
$$

$$
= 1 - \frac{\delta |\mathscr{K}|}{dL_{\max}^2 \log L_{\max}} \geq 1 - \delta.
$$

Let $\widehat{L}$ be the output of Algorithm 1. We show that, with probability $1 - \delta$, $\widehat{L}$ is $(\rho, \sqrt{d})$-anp of $L$. In fact, suppose $\widehat{L}$ is not $(\rho, \sqrt{d})$-anp. We note that $\widehat{L} < L$. There exists $s \in \{0, \ldots, \widehat{L} - 1\}$ and $t_1, t_2 \in \mathbb{Z}_{>0}$,

$$
\| f^{s + \widehat{L} t_1}(\theta) - f^{s + \widehat{L} t_2}(\theta) \|_{\mathbb{R}^d} > \rho + 2\sqrt{d}\mu.
$$

Let $m' \in \operatorname{argmax}_{m=1,\ldots,d} \left| f^{s + \widehat{L} t_1}(\theta)^\top \mathbf{u}_m - f^{s + \widehat{L} t_2}(\theta)^\top \mathbf{u}_m \right|$. Put $a_t := f^{s + \widehat{L} t}(\theta)^\top \mathbf{u}_{m'}$. Then, for any $t, t' \in \mathbb{Z}_{>0}$, we have

$$
|a_t - a_{t'}| \geq |a_{t_1} - a_{t_2}| - 2\mu
$$

$$
\geq \frac{\| f^{s + \widehat{L} t_1}(\theta) - f^{s + \widehat{L} t_2}(\theta) \|_{\mathbb{R}^d}}{\sqrt{d}} - 2\mu
$$

$$
> \frac{\rho}{\sqrt{d}}.
$$

Thus, by definition, we have

$$
\sigma_{L/\widehat{L}}((a_t)_t) \geq \frac{\rho}{2\sqrt{dL/\widehat{L}}} \geq \frac{\rho}{2\sqrt{dL_{\max}}}.
$$

Let $\sigma_0 = \rho/2\sqrt{dL_{\max}}$ and $\xi := \gamma^{-1}/3\sqrt{L_{\max}}$. Then, $\mu/(\gamma\xi) < \sigma_0 \leq \sigma_{L/\widehat{L}}((a_t)_t)$, and

$$
\begin{aligned}
&\frac{8L_{\max}R^2\log(4/\delta)}{\sigma_0^2(\xi-\lambda)^2} + \frac{36L_{\max}^{5/2}\sup_{t\geq 1}|a_t|}{\sigma_0(\xi-\lambda)} \\
&\leq \frac{32\xi^{-2}dL_{\max}^2R^2\log(4/\delta)}{\rho^2(1-r)^2} + \frac{72\xi^{-1}\sqrt{d}L_{\max}^3\sup_{t\geq 1}|a_t|}{\rho(1-r)} \\
&\leq \frac{72dL_{\max}^2R^2\log(4/\delta)}{\rho^2(1-r)^2} + \frac{108\sqrt{d}L_{\max}^3\sup_{t\geq 1}|a_t|}{\rho(1-r)}.
\end{aligned}
$$

The last inequality follows from $\xi \geq 2/3$. Thus, by Proposition 5.2, the algorithm finds $\beta > \widehat{L}$ in $m'$-th loop, and the output becomes an integer larger than $\widehat{L}$, which is contradiction. $\qquad\square$

## C  Proof of Theorem 5.8

Here, we provide the proof of Theorem 5.8.

### C.1  Proof

Let

$$
K := (M\theta, \ldots, M^d\theta) : \mathbb{C}^d \to W,
$$
$$
E_s(N) := \mathscr{W}\left((E_{s,j})_{j=0}^{N-1}\right) : \mathbb{C}^d \to \mathbb{C}^d.
$$

For a linear map $\mathcal{M} : W \to W$, we define linear maps:

$$
X(\mathcal{M}) := \begin{pmatrix} \tilde{x}_1^\top \mathcal{M}^1 \\ \tilde{x}_2^\top \mathcal{M}^{2d+1} \\ \vdots \\ \tilde{x}_d^\top \mathcal{M}^{2(d-1)d+1} \end{pmatrix} : W \to \mathbb{C}^d,
$$
$$
Q_N(\mathcal{M}) := \mathscr{W}\left((\mathcal{M}^{2d^2 j})_{j=0}^{N-1}\right) : W \to W.
$$

For $s = 0, 1$, we define linear maps on $\mathbb{C}^d$ by

$$
A_s(N; \mathcal{M}) := X(\mathcal{M})Q_N(\mathcal{M})\mathcal{M}^{sd+N-1}K + E_s(N),
$$
$$
B_s(N; \mathcal{M}) := X(\mathcal{M})Q_N(\mathcal{M})\mathcal{M}^{sd+N-1}K.
$$

We note that $A_s(N; M_\theta)$ is identical to $A_s(N)$ defined in (5.7). We impose the following assumption on $X(M_\theta)$:

**Assumption 5.** *The kernel of the linear map $X(M_1)$ is the same as $\mathcal{N}(M_1)$.*

Note that this assumption holds with probability 1 if we randomly choose $\tilde{x}_1, \ldots, \tilde{x}_d$ (see Lemma C.7).

The following *isomorphicity maintenance lemma* provides an explicit description of $B_0(N; M_1)^\dagger$.

**Lemma C.1** (Isomorphicity maintenance lemma)**.** *Suppose Assumption 5 holds. Assume $N \geq 16L^2$. Let*

$$
U_1 := \mathscr{I}(B_0(N; M_1)) \subset \mathbb{C}^d,
$$
$$
U_2 := \mathscr{N}(B_0(N; M_1)) \subset \mathbb{C}^d,
$$
$$
U_3 := \mathscr{I}(M_1) = W_{=1} \subset W.
$$

*Let $i : U_1 \to \mathbb{C}^d$ be the inclusion map and $p : \mathbb{C}^d \to U_2^\perp$ the orthogonal projection. Then, restriction of $X(M_1)$ (resp. $M_1K$) to $U_3$ (resp. $U_2^\perp$) induces an isomorphism onto $U_1$ (resp. $U_3$). If we denote the isomorphism by $\tilde{X}$ (resp, $\tilde{K}$). Then, $B_0(N; M_1)^\dagger$ is given by*

$$
B_0(N; M_1)^\dagger = p^* \tilde{K}^{-1} M_1|_{U_3}^{2-N} Q_N(M_1)|_{U_3}^{-1} \tilde{X}^{-1} i^*.
$$

*Proof.* Since we have $\mathscr{N}(M_1) = \mathscr{N}(M_1^{r+1})$ for all $r \geq 0$ and Assumption 5, surjectivity of $K$ by Proposition C.5, and bijectivity of $Q_N(M_1)$ on $U_3$ by Proposition 5.7, we have

$$U_1 = \mathscr{I}(X(M_1)|_{U_3}),$$
$$U_2 = \mathscr{N}(M_1 K).$$

Thus, the restriction of $X(M_1)$ (resp. $K$) to $U_3$ (resp. $U_2^{\perp}$) induces an isomorphism onto $U_1$ (resp. $U_3$). Let us denote the isomorphism by $\tilde{X}$ (resp. $\tilde{K}$). Then, the last statement follows from Proposition C.4. □

**Lemma C.2.** *Assume $N \geq 16L^2$. Let*

$$A := B_1(N; M_1)B_0(N; M_1)^{\dagger}.$$

*Then, $A$ is independent of $N$ and its eigenvalues are zeros except for $(\theta, d)$-distinct eigenvalues of $M$.*

*Proof.* We use the notation as in Lemma C.1. By Lemma C.1, we obtain

$$B_1(N; M_1)B_0(N; M_1)^{\dagger} = i\tilde{X}M_1^d\tilde{X}^{-1}i^*,$$

which is independent of $N$ and its eigenvalues are zeros except for $(\theta, d)$-distinct eigenvalues of $M$. □

The following result will be used in the proof of Theorem 5.8 (but not essential).

**Lemma C.3.** *We have*

$$\|X(M_1)\| \leq \kappa\sqrt{d},$$
$$\|X(M_{<1})\| \leq \kappa(d+1)2^d.$$

*Proof.* Considering the Jordan normal form, the first inequality is obvious. As for the second inequality, we define $J \in \mathbb{R}^{d \times d}$ by the nilpotent matrix

$$J := \begin{pmatrix} 0 & 1 & & O \\ & \ddots & \ddots & \\ & & & 1 \\ O & & & 0 \end{pmatrix}.$$

Then, we have

$$\kappa^{-1}\|X(M_1)\| \leq \sum_{r=1}^{d} \|(I+J)\|_1$$

$$= \sum_{r=1}^{d}\sum_{j=0}^{d} \binom{r}{j}(d-j)$$

$$= \sum_{j=0}^{d}(d-j)\sum_{r=1}^{d}\binom{r}{j}$$

$$= d + \sum_{j=1}^{d}(d-j)\sum_{r=1}^{d}\binom{r}{j}$$

$$= d + \sum_{j=1}^{d}(d-j)\binom{r+1}{j+1}$$

$$= d + (d+1)2^d - (d+1)d - (d+1)$$

$$< (d+1)2^d.$$

□

*Proof of Theorem 5.8.* Let $A$ be the matrix introduced in Lemma C.2. Let

$$\gamma(N) := \frac{\sqrt{4d^2 R^2 \log(4d^2/\delta)} + 1}{\sqrt{N}}.$$

Let $M_1$ and $M_{<1}$ be matrices as in Corollary 4.3. Then, by Proposition 4.2, we see that

$$A_s(N, M) = A_s(N, M_1) + A_s(N, M_{<1}),$$
$$B_s(N, M) = B_s(N, M_1) + B_s(N, M_{<1}).$$

We denote by $\hat{A}_s(N; M)$ (resp, $\hat{B}_s(N; M)$) the low rank approximation via the singular value threshold $\gamma(N)$ (see Definition 5.4). By direct computations, we have

$$\|A - A_1(N; M)\hat{A}_0(N; M)^\dagger\|$$
$$\leq \|A_1(N; M)\| \cdot \|\hat{A}_0(N; M)^\dagger - B_0(N; M_1)^\dagger\| + \|A_1(N; M) - B_1(N; M_1)\| \cdot \|B_0(N; M_1)^\dagger\|$$
$$\leq (\|B_1(N; M_1)\| + \|B_1(N; M_{<1})\| + \|E_1(N)\|) \cdot \|\hat{A}_0(N; M)^\dagger - B_0(N; M_1)^\dagger\|$$
$$\quad + \|B_1(N; M_{<1}) + E_1(N)\| \cdot \|B_0(N; M_1)^\dagger\|.$$

By Proposition 5.7, for $s = 0, 1$ and for $N \geq \max\{2d, 16L^2\}$, we have

$$\|B_s(N; M_1)\| \leq \|X(M_1)\| \cdot \|M_1^{N+sd-1} Q_N(M_1)\| \cdot \|K\|$$
$$\leq \kappa C_{\mathrm{ws}}(L)\|X(M_1)\|\|K\|$$
$$\leq Bd\kappa^2 C_{\mathrm{ws}}(L)$$
$$\|B_s(N; M_{<1})\| \leq \|X(M_{<1})\| \cdot \|M_{<1}^{N+sd-1} Q_N(M_{<1})\| \cdot \|K\|$$
$$\leq \|X(M_{<1})\| \cdot \|K\| \cdot \frac{d^2 \kappa e^{-\Delta(N+sd-d)}}{N\Delta^{d-1}}$$
$$\leq B\kappa\sqrt{d}(d+1)2^d \cdot \frac{d^2 \kappa e^{-\Delta(N+sd-d)}}{N\Delta^{d-1}}$$
$$\leq \frac{B\kappa^2 e^{d+6-\Delta(N+sd-d)}}{N\Delta^{d-1}},$$

where we used $\|K\| \leq \sqrt{d}B$ (Assumption 3), and $\|X(M_{<1})\| \leq \kappa(d+1)2^d$ (Lemma C.3), and $\sqrt{d}(d+1)d^2 2^d < e^{d+6}$. By Lemma C.1 with Proposition 5.7, we see that

$$\|B_0(N; M_1)^\dagger\| \leq \kappa C_{\mathrm{ws}}(L)\|\tilde{X}^{-1}\| \cdot \|\tilde{K}^{-1}\|.$$

By using Lemma D.1 and union bounds, we obtain

$$\max(\|E_0(N)\|_F, \|E_1(N)\|_F) \leq \gamma(N) - \frac{1}{\sqrt{N}},$$

with probability at least $1 - \delta$. Assume that

$$N \geq \frac{-(d-1)\log\Delta}{\Delta} + \frac{\log(B\kappa^2) + d + 6}{\Delta} + d.$$

Then, we see that $\|B_s(N; M_{<1})\| \leq 1/N$. Thus, by Lemma C.6, with probability at least $1 - \delta$, we have

$$\|\hat{A}_0(N; M)^\dagger - B_0(N; M_1)^\dagger\| \leq 8(\|E_0(N)\| + \|B_0(N; M_{<1})\|) \cdot \|B_0(N, M_1)^\dagger\|^2(\sqrt{d} + 1)$$
$$\leq 8\gamma(N) \cdot \|B_0(N, M_1)^\dagger\|^2(\sqrt{d} + 1)$$
$$\leq 8(1 + \sqrt{d})\kappa^2 C_{\mathrm{ws}}(L)^2\|\tilde{X}^{-1}\|^2 \cdot \|\tilde{K}^{-1}\|^2\gamma(N).$$

Therefore, there exists $C > 0$ depending on $\|X(M)\|$, $\|K\|$, $\kappa$, $C_{\mathrm{ws}}(L)$, $\|\tilde{X}^{-1}\|$, $\|\tilde{K}^{-1}\|$, and $d$ such that

$$\|A - A_1(N; M)\hat{A}_0(N; M)^\dagger\| < C\left(\frac{R^2\left(\log\left(1/\delta\right) + 1\right) + 1}{\sqrt{N}}\right),$$

with probability at least $1 - \delta$. $\qquad\square$

## C.2 Miscellaneous

We provide an expression of Moore-Penrose pseudo inverse:

**Proposition C.4.** *Let $A : \mathbb{C}^m \to \mathbb{C}^n$ be a linear map. Let $i : \mathscr{I}(A) \to \mathbb{C}^n$ be the inclusion map and let $p : \mathbb{C}^m \to \mathscr{N}(A)^\perp$ be the orthogonal projection. Let $\tilde{A} := A|_{\mathscr{N}(A)} : \mathscr{N}(A)^\perp \to \mathscr{I}(A)$ be an isomorphism. Then, the Moore-Penrose pseudo inverse $A^\dagger$ coincides with $i^* \tilde{A}^{-1} p^*$.*

*Proof.* Let $B := p^* \tilde{A}^{-1} i^*$. We remark that $A = i\tilde{A}p$, $i^*i = \mathrm{id}, pp^* = \mathrm{id}$, we see that $ABA = A$, $BAB = B$, $(AB)^* = AB$, and $BA = (BA)^*$. By the uniqueness of Moore-Penrose pseudo inverse, $B = A^\dagger$. $\qquad\square$

**Proposition C.5.** *Let $A \in \mathbb{C}^{d \times d}$ be a matrix and let $v \in \mathbb{C}^d$ be a vector. Let $V \subset \mathbb{C}^d$ be a linear subspace generated by $\{A^j v\}_{j=1}^\infty$. Then, $V = \mathscr{I}\left((Av, A^2 v, \ldots, A^d v)\right)$.*

*Proof.* Put $W = \mathscr{I}\left((Av, A^2 v, \ldots, A^d v)\right)$. The inclusion $W \subset V$ is obvious, we prove the opposite inclusion. It suffices to show that $A^j v \in W$ for any positive integer $j > d$. By the Cayley-Hamilton theorem, $A^d = \sum_{j=1}^d c_j A^{d-j}$ for some $c_j \in \mathbb{C}$. Thus, by induction $A^j$ is a linear combination of $A, A^2, \ldots, A^d$, namely, $A^j v \in W$. $\qquad\square$

We give several lemmas here.

**Lemma C.6** (Perturbation bounds of the Moore-Penrose inverse)**.** *Suppose $A \in \mathbb{C}^{d \times d}$ is a matrix. Let $E \in \mathbb{C}^{d \times d}$ be a matrix satisfying*

$$\exists C > 0, \quad \forall N \in \mathbb{Z}_{>0}, \quad \|E\| \leq C \frac{1}{\sqrt{N}},$$

*and let $\hat{A}_E \in \mathbb{C}^{d \times d}$ be the low rank approximation of $A+E$ via SVD with the singular value threshold $C/\sqrt{N}$. Then, we obtain*

$$\left\| A^\dagger - \hat{A}_E^\dagger \right\| \leq \frac{8C \|A^\dagger\|^2 \left(\sqrt{d}+1\right)}{\sqrt{N}}.$$

*Proof.* Let $\sigma_{\min} = \|A^\dagger\|^{-1}$ be the minimal singular value of $A$. From (Meng & Zheng, 2010, Theorem 1.1) (or originally Wedin (1973)) and from the fact

$$\left\| \hat{A}_E - A \right\| \leq \|E\| + \frac{\sqrt{d} C}{\sqrt{N}},$$

we obtain

$$\left\| A^\dagger - \hat{A}_E^\dagger \right\| \leq \frac{1+\sqrt{5}}{2} \max\left\{ \left\|A^\dagger\right\|^2, \left\|\hat{A}_E^\dagger\right\|^2 \right\} \left(\sqrt{d}+1\right) \frac{C}{\sqrt{N}}.$$

For $N$ such that $1 \leq N \leq 4C^2/\sigma_{\min}^2$, we have $\left\|\hat{A}_E^\dagger\right\| \leq \sqrt{N}/C \leq 2/\sigma_{\min}$. Suppose singular values $\sigma_k$, $k \in [d]$, of $A$, and $\hat{\sigma}_k$, $k \in [d]$, of $A + E$ are sorted in descending order. Then, it holds that

$$|\sigma_k - \hat{\sigma}_k| \leq \|E\|.$$

Therefore, for $N > 4C^2/\sigma_{\min}^2$, the minimum singular value $\hat{\sigma}_d$ is greater than $\sigma_{\min}/2$. In this case, $\hat{A}_E = A + E$, and it follows that $\left\|\hat{A}_E^\dagger\right\| \leq 2/\sigma_{\min}$. Hence, for all $N \geq 1$, we obtain

$$\left\|\hat{A}_E^\dagger\right\| \leq \frac{2}{\sigma_{\min}},$$

from which, it follows that

$$\left\| A^\dagger - \hat{A}_E^\dagger \right\| \leq \frac{2C\left(1+\sqrt{5}\right)\left(\sqrt{d}+1\right)}{\sigma_{\min}^2 \sqrt{N}} \leq \frac{8C\|A^\dagger\|^2\left(\sqrt{d}+1\right)}{\sqrt{N}}.$$

$\square$

**Lemma C.7** (Null space of random matrix)**.** *Suppose $M_k \in \mathbb{R}^{d \times d}$, $k \in [d]$, have the same null space. Then, the null space of*

$$X := \begin{pmatrix} x_1^\top M_1 \\ x_2^\top M_2 \\ \vdots \\ x_d^\top M_d \end{pmatrix},$$

*where $x_k$, $k \in [d]$, are unit vectors independently drawn from the uniform distribution over the unit hypersphere, is the same as those of $M_k$ with probability one.*

*Proof.* Given any $k-1$ dimensional linear subspace in $\mathcal{N}(M_k)^\perp$ for any $k \in [d]$, it holds that the probability that $x_k^\top M_k$ lies on that space is zero. Therefore, by union bound, and by the fact that the row space is the orthogonal complement of the null space, we obtain the result. $\square$

## D  Azuma-Hoeffding inequality for exponential sum

**Lemma D.1.** *Let $\{X_j\}_{j=1}^n$ for $n \in \mathbb{Z}_{>0}$ be sub-Gaussian martingale difference with variance proxy $R^2$ and a filtration $\{\mathcal{F}_j\}$. Also, let $\{a_j\} \subset \mathbb{C}$ be a sequence of complex numbers satisfying $|a_j| \leq 1$ for all $j \in [n]$. Then, the followings hold, where $*[\cdot]$ stands for $\mathfrak{Re}[\cdot]$ or $\mathfrak{Im}[\cdot]$.*

$$\Pr\left[\frac{1}{n}\left|*\left[\sum_{j=1}^n a_j X_j\right]\right| \leq \sqrt{\frac{2R^2 \log(2/\delta)}{n}}\right] \geq 1 - \delta,$$

$$\Pr\left[\frac{1}{n}\left|\sum_{j=1}^n a_j X_j\right| \leq \sqrt{\frac{4R^2 \log(4/\delta)}{n}}\right] \geq 1 - \delta.$$

*Proof.* For a filtration $\{\mathcal{F}_i\}_{i \leq n}$, we have

$$\mathbb{E}\left[e^{\lambda \mathfrak{Re}\left[\sum_{j=1}^n a_j X_j\right]}\right] \leq \mathbb{E}\left[e^{\lambda \mathfrak{Re}\left[\sum_{j=1}^{n-1} a_j X_j\right]}\mathbb{E}\left[e^{\lambda \mathfrak{Re}[a_n X_n]}\big|\mathcal{F}_{n-1}\right]\right] \leq e^{\frac{\lambda^2 R^2}{2}}\mathbb{E}\left[e^{\lambda \mathfrak{Re}\left[\sum_{j=1}^{n-1} a_j X_j\right]}\right],$$

where the first inequality follows from the assumption of filtration, and the second inequality follows from

$$\mathbb{E}\left[e^{\lambda \mathfrak{Re}[a_n X_n]}\big|\mathcal{F}_{n-1}\right] = \mathbb{E}\left[e^{\lambda X_n \mathfrak{Re}[a_n]}\big|\mathcal{F}_{n-1}\right] \leq e^{\frac{\lambda^2 R^2}{2}}.$$

By induction, we obtain

$$\mathbb{E}\left[e^{\lambda \mathfrak{Re}\left[\sum_{j=1}^n a_j X_j\right]}\right] \leq e^{\frac{n\lambda^2 R^2}{2}}.$$

By using Markov inequality and the union bound, it follows that

$$\Pr\left[\frac{1}{n}\left|\mathfrak{Re}\left[\sum_{j=1}^n a_j X_j\right]\right| > \epsilon\right] \leq 2e^{-\frac{n\epsilon^2}{2R^2}}.$$

Similarly, we have

$$\Pr\left[\frac{1}{n}\left|\mathfrak{Im}\left[\sum_{j=1}^{n}a_jX_j\right]\right| > \epsilon\right] \le 2e^{-\frac{n\epsilon^2}{2R^2}}.$$

Therefore, we obtain

$$\Pr\left[\frac{1}{n}\left|\sum_{j=1}^{n}a_jX_j\right| \le \sqrt{\frac{4R^2\log\left(4/\delta\right)}{n}}\right]$$

$$\ge \Pr\left[\frac{1}{n}\left|\mathfrak{Re}\left[\sum_{j=1}^{n}a_jX_j\right]\right| \le \sqrt{\frac{2R^2\log\left(4/\delta\right)}{n}}\right]$$

$$+ \Pr\left[\frac{1}{n}\left|\mathfrak{Im}\left[\sum_{j=1}^{n}a_jX_j\right]\right| \le \sqrt{\frac{2R^2\log\left(4/\delta\right)}{n}}\right] - 1$$

$$\ge \left(1 - \frac{\delta}{2}\right) + \left(1 - \frac{\delta}{2}\right) - 1 \ge 1 - \delta,$$

where the first inequality follows from the Fréchet inequality. $\square$

## E   Applications to bandit problems

We briefly cover the applicability of our proposed algorithms to bandit problems (e.g., regret minimization) which is mentioned in the introduction.

A naive approach is an explore-then-commit type algorithm (cf. Robbins (1952); Anscombe (1963)). One employs our algorithm to estimate a nearly period, followed by a certain periodic bandit algorithm such as the work in Cai et al. (2021) to obtain an asymptotic order of regret. Caveat here is, because our estimate is only an aliquot nearly period, one may need to take into account the regret caused by this misspecification when running bandit algorithms (e.g., $\rho$ and $\mu$ may lead to (small) linear regret). Avoiding this small linear regret would require the system to be 0-nearly periodic and that there exists a sufficiently large *gap* ensuring $\mu$-nearly period with sufficiently small $\mu$ implies 0-nearly period.

If one aims at designing an anytime algorithm, the straightforward application of our algorithms may not give near optimal asymptotic rate of *expected* regret because the failure probability of periodic structure estimations cannot be adjusted later. To remedy this, one can employ our algorithm repetitively, and gradually increase the span of such procedure. Importantly, samples from separated spans can contribute to the estimate together when the *surplus* beyond a multiple of period is properly dealt with. Since failure probability decreases exponentially with respect to sample size, we conjecture that increasing the span for bandit algorithm by a certain order will lead to the same rate (up to logarithm) of *expected* regret of the adopted bandit algorithm.

## F   Simulation setups and results

Throughout, we used the following version of Julia Bezanson et al. (2017); for each experiment, the running time was less than a few minutes.

```
Julia Version 1.6.3
Platform Info:
OS: Linux (x86_64-pc-linux-gnu)
CPU: Intel(R) Core(TM) i7-6850K CPU @ 3.60GHz
WORD_SIZE: 64
LIBM: libopenlibm
```

**Table 3:** Hyperparameters used for period estimation of LifeGame.

| LifeGame hyperparameter | Value | Algorithm hyperparameter | Value |
|---|---|---|---|
| height | 12 | accuracy for estimation $\rho$ | 0.98 |
| width | 12 | failure probability bound $\delta$ | 0.2 |
| observed dimension | 5 | maximum possible period $L_{\max}$ | 10 |
| observation noise proxy $R$ | 0.3 | | |
| ball radius $B$ | $\sqrt{5}$ | | |

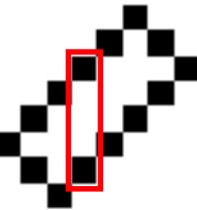

**Figure 8:** The area we focus on for the cellular automata experiment.

```
LLVM: libLLVM-11.0.1 (ORCJIT, broadwell)
Environment:
JULIA_NUM_THREADS = 12
```

We also used some tools and functionalities of Lyceum Summers et al. (2020). The licenses of Julia and Lyceum are [The MIT License; Copyright (c) 2009-2021: Jeff Bezanson, Stefan Karpinski, Viral B. Shah, and other contributors: https://github.com/JuliaLang/julia/contributors], and [The MIT License; Copyright (c) 2019 Colin Summers, The Contributors of Lyceum], respectively.

In this section, we provide simulation setups, including the details of parameter settings.

### F.1 Period estimation: LifeGame

The hyperparameters of LifeGame environment and the algorithm are summarized in Table 3. Note $\mu = 0$ because it is a periodic transition. Here, we used $12 \times 12$ blocks of cells and we focused on the five blocks surrounded by the red rectangle in Figure 8. The transition rule is given by

1. If the cell is alive and two or three of its surrounding eight cells are alive, then the cell remains alive.

2. If the cell is alive and more than three or less than two of its surrounding eight cells are alive, then the cell dies.

3. If the cell is dead and exactly three of its surrounding eight cells are alive, then the cell is revived.

### F.2 Period estimation: Simple $\mu$-nearly periodic system

The dynamical system

$$r_{t+1} = \mu \left( \alpha \frac{r_t - 1}{\mu} - \lceil \alpha \frac{r_t - 1}{\mu} \rceil \right) + 1, \qquad \theta_{t+1} = \theta_t + \frac{2\pi}{L},$$

is $\mu$-nearly periodic. See Figure 9 for the illustrations when $\mu = 0.2$, $L = 5$, $\alpha = \pi$. It is observed that there are five *clusters*. We mention that this system is not exactly periodic. The hyperparameters of this system and the algorithm are summarized in Table 4.

**Table 4:** Hyperparameters used for $\mu$-nearly periodic system.

| System hyperparameter | Value | Algorithm hyperparameter | Value |
|---|---|---|---|
| dimension | 2 | accuracy for estimation $\rho$ | 0.3 |
| $\mu$ | 0.001 | failure probability bound $\delta$ | 0.2 |
| $\alpha$ | $\pi$ | maximum possible nearly period $L_{\max}$ | 8 |
| observation noise proxy $R$ | 0.3 | | |
| ball radius $B$ | 2 | | |

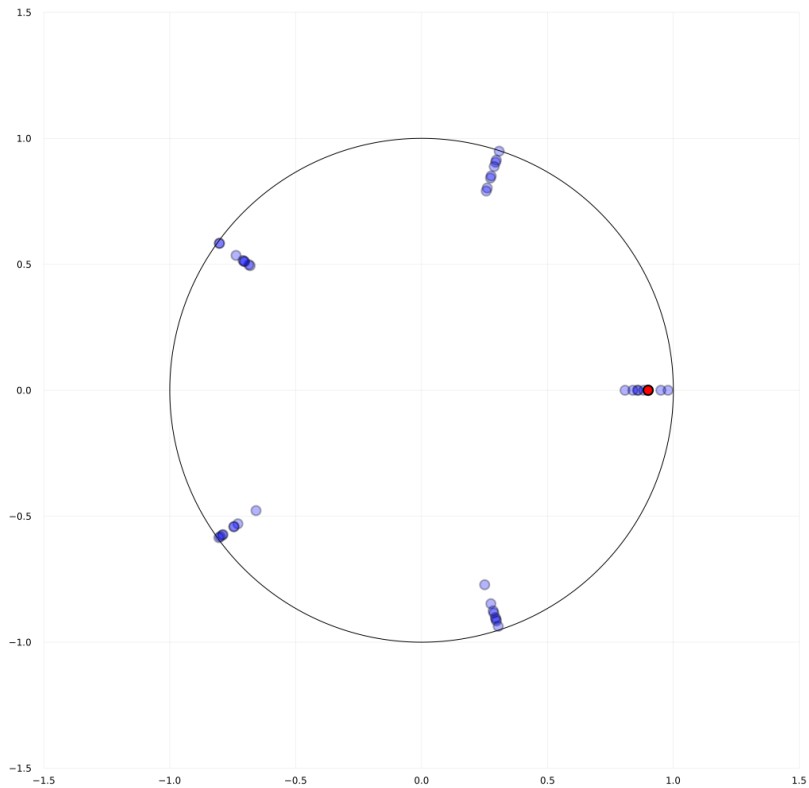

**Figure 9:** An example of $\mu$-nearly periodic system.

## F.3  Eigenvalue estimation

We used the matrix $M$ given by

$$
M := \begin{pmatrix} 0 & 0 & 0 & 1 & 0 \\ 0 & 1 & 0 & 0 & 0 \\ 1 & 0 & 0 & 0 & 0 \\ 0 & 0 & 1 & 0 & 0 \\ 0 & 0 & 0 & 0 & 0.7 \end{pmatrix}.
$$

The first $4 \times 4$ block matrix is for permutation. After $N$ steps, it is expected that the last dimension shrinks so that the system becomes nearly periodic. It follows that $4! = 24$ is a multiple of the length $L$. Eigenvalues of $M^5$ are given by $1.000$, $1.000$, $-0.500 - 0.866i$, $-0.500 + 0.866i$, $0.168$, and the $(\theta_0, 5)$-distinct eigenvalues are $1.000$, $-0.500 - 0.866i$, $-0.500 + 0.866i$.

The hyperparameters of the environment and the algorithm are summarized in Table 5. Note we don't necessarily need $\kappa$, $B$, and $\Delta$ to run the algorithm as long as the effective sample size is sufficiently large; we used the values (satisfying the conditions) in Table 5 for simplicity.

**Table 5:** Hyperparameters used for eigenvalue estimation.

| Hyperparameter | Value | Hyperparameter | Value |
|---|---|---|---|
| $\kappa$ | 6 | a nearly period $L$ | 24 |
| $\Delta$ | 0.1 | failure probability bound $\delta$ | 0.2 |
| dimension | 5 | observation noise proxy $R$ | 0.3 |
| ball radius $B$ | 1 | | |

### F.4 Realistic simulation

The hyperparameters used for period estimation that outputs an invalid estimate are summarized in Table 6 while those for a reasonable output are summarized in Table 7. Also, the hyperparameters used for the eigenvalue estimation are summarized in Table 8. Note for all of them, we used the fixed number of sample sizes.

**Table 6:** Hyperparameters used for worm simulation that outputs incorrect value.

| System hyperparameter | Value | Algorithm hyperparameter | Value |
|---|---|---|---|
| dimension | 36 | accuracy for estimation $\rho$ | 1.0 |
| sample number $T_p$ | $3 \times 10^5$ | maximum possible nearly period $L_{\max}$ | 25 |
| observation noise proxy $R$ | 0 | | |

**Table 7:** Hyperparameters used for worm simulation that outputs reasonable value.

| System hyperparameter | Value | Algorithm hyperparameter | Value |
|---|---|---|---|
| dimension | 36 | accuracy for estimation $\rho$ | 20.0 |
| sample number $T_p$ | $3 \times 10^5$ | maximum possible nearly period $L_{\max}$ | 25 |
| observation noise proxy $R$ | 0 | | |

**Table 8:** Hyperparameters used for eigenvalue estimation of worm simulation.

| Hyperparameter | Value | Hyperparameter | Value |
|---|---|---|---|
| sample number $N$ | $10^3, 10^4, 10^5$ | failure probability bound $\delta$ | 0.2 |
| dimension | 36 | observation noise proxy $R$ | 1.0 |
| a nearly period $L$ | 20 | | |

