# OpenReview forum: "Dynamic Structure Estimation from Bandit Feedback using Nonvanishing Exponential Sums"
_TMLR — Accepted by TMLR_

### Review · Reviewer_nJrm · 2024-03-13

**Summary Of Contributions:**

The paper studies the problem of estimating periodic information, such as (effective) period and selected subset of eigenvalues, about dynamical systems from bandit feedback under sub-Gaussian noises.

The primary technique leverages some analytical results on exponential sums, which reduce noises while retaining sufficient information about the underlying system. For a sufficiently large sample and properly chosen parameters (and some additional assumptions about the system), the results are two-fold: 1) One can estimate the (effective) period of the denoised output sequence; 2) For a linear system, one can further approximately recover some of the eigenvalues.

The paper complements the theoretical claims and theorems with a toy experiment/example from the famous LifeGame of Conway. With properly chosen parameters, the synthetic results are broadly consistent and accurate.

**Audience:**

Yes

**Broader Impact Concerns:**

The submission is mainly theoretical in the current format, demonstrating its application via a toy problem. I do not have immediate concerns about the ethical implications of the work that would require adding a "Broader Impact Statement."

**Claims And Evidence:**

Yes

**Requested Changes:**

Please refer to the Weaknesses subsection under "Strengths and Weaknesses." Specifically, I would suggest the authors to:
1. Reduce the redundancy in the Introduction and Related Work, and, if possible, discuss how the techniques and results differ from the closest research work(s).
2. Justify the quality and goodness of the results and guide parameter selection in practical applications (with reasonable justification).
3. Conduct some experiments on real-world data to mitigate the discrepancy between problem-motivating applications and the experiments in the paper.

**Strengths And Weaknesses:**

On top of some of the strengths mentioned in the "Summary of Contributions":
1. Both Period/Eigenvalue estimation results are associated with non-asymptotic bounds. The proof logic and the introduction of notations and definitions are also smoothly presented.
2. The use of exponential sum, particularly Weyl sum, is likely of independent interest and connects practical applications and well-known analytical number theory results. Proposition 4.7 simultaneously ensures the protection of eigenvalue information and the cancellation of noises.
3. The experimental results look good as the algorithm achieved zero error on Period Estimation and near-zero error on Eigenvalue estimation. Note that the period value is an integer, so zero error may be expected for other algorithms. However, eigenvalues are Real values, and algorithms are unlikely to achieve zero error in general settings.

Weaknesses:
1. The Related Work and Introduction sections introduce many relevant works and seem redundant, especially since the paper does not dive deep into any research closely related to its own.
2. Theorems 4.3 and 4.8 present some sample/observation size bounds for the underlying problems but do not justify their optimality. While the authors mentioned that this could be some future research direction, it is a common scenario where the difficulty of a statistical estimation problem decreases significantly for sufficiently large samples. For this very reason, researchers often aim to determine the min-max sample complexities or regrets. I suggest the authors consider providing some justifications for the quality and goodness of their results. Similarly, it's unclear how to choose the parameters for practical applications.
3. The introduction motivates the problem through many vital applications in control, economics, statistical machine learning, and biological/earthquake/chemical/asteroseismic analysis. However, the experiment section presents only a toy example with a period 8. This seems to be a significant discrepancy. Therefore, I would suggest the authors conduct some experiments on real-world data to close this gap.

---

> ### Author Response · Authors · 2024-04-18
> **Thank you so much for your suggestions**
>
> We sincerely thank you for kindly taking your precious time reviewing our paper.  Your comments have helped us further strengthen our work.
> Below, we answered to your questions and we revised the manuscript accordingly.
>
> >The Related Work and Introduction sections introduce many relevant works and seem redundant, especially since the paper does not dive deep into any research closely related to its own.
>
> Thank you for the comment; we made the introduction section more concise by focusing on what our work is doing.  Then, following a comment from the reviewer **okMR**, we added a section that summarizes the (rather broad) motivation behind our work.  Moreover, we modified the related work section to better place our work in the relations to existing lines of work.
>
> >Theorems 4.3 and 4.8 present some sample/observation size bounds for the underlying problems but do not justify their optimality. While the authors mentioned that this could be some future research direction, it is a common scenario where the difficulty of a statistical estimation problem decreases significantly for sufficiently large samples. For this very reason, researchers often aim to determine the min-max sample complexities or regrets. I suggest the authors consider providing some justifications for the quality and goodness of their results.
>
> Thank you for the comment; we extended the corresponding paragraph in the discussion section.  As we mentioned in the manuscript, we first kindly remind you that the current assumptions (i.e., without boundedness assumption on the set of arm $D$) preclude the discussions on the lower bound of minimally required samples for the estimations.  Therefore, to discuss the lower bound, we need to consider proper assumptions here as well.
>
> As such, instead of arguing the tightness, to justify the complexity of our algorithms, we hence mention that the very naive approach for (nearly) period estimation would cost the order of $L_{\rm max}!$ sample complexity.  To see this, suppose we know the maximum possible (nearly) period $L_{\rm max}$ then it follows that $L_{\rm max}!$ is a true (nearly) period as a multiple of the fundamental (nearly) period.  As such, one could employ the concentration of measures to average out the sample .  With sufficiently many sequences of length $L_{\rm max}!$ depending on the rate of concentration, one could approximately reconstruct the original (noiseless) sequence of length $L_{\rm max}!$ from which one could estimate a desired $L\leq L_{\rm max}$ which is an aliquot nearly period.
>
> In the future, lower bound of the minimally required samples may be discussed by using similar arguments to the best arm identification problems which typically employ similar theoretical results; however, our problems consider dynamical systems, and we have no clue on potential technical tools to deal with the arguments at this point.  In that sense, we would like to mention that discussing such bounds should be nontrivial in our rather novel problem setups.
>
> >Similarly, it's unclear how to choose the parameters for practical applications.
>
> Thank you for the comment; with regard to the discussions on the parameter selection, we would like to ask you to read the response below to your comment on applications to real data.  We briefly repeat the content; the output of our algorithms could become contradicting to the parameter setting when the hypeparameters are wrong or do not reflect the target properties.  Therefore, it would make sense to, for example, increase $L_{\rm max}$, $R$ etc. until you get reasonable output.  We added this discussion in the simulated experiments section and modified the discussion section as well.

---

> > ### Author Response · Authors · 2024-04-18
> > **Additional responses**
> >
> > >The introduction motivates the problem through many vital applications in control, economics, statistical machine learning, and biological/earthquake/chemical/asteroseismic analysis. However, the experiment section presents only a toy example with a period 8. This seems to be a significant discrepancy. Therefore, I would suggest the authors conduct some experiments on real-world data to close this gap.
> >
> > Thank you for your suggestion; although it is not the ''real'' data, we used OpenWorm simulations to generate a realistic but simulated motion of a worm which is known to contain some periodicity and conducted an additional experiment.  It is added to the simulated experiments section.  While we would like to ask you to look at the section, we briefly state the results.
> >
> > We ran more than around $10$ hours for simulations to get video data, and we resized each image in the video to $6\times6$ (i.e., $d=36$), and downsampled it by extracting one per eight images.  Also, we repeat the same video in the way that the entire video still shows smooth flow (i.e., cut the video so that the end image is similar to the image in the beginning, and concatenate the same videos) so that we can sample a large number of data.  Although the motion is not exactly periodic, we roughly see there exist six ''periods'' inside the sequence of images of length $120$.  With certain choice of hyperparameters, the period estimation algorithm outputs $L=40$ even though $L_{\rm max}$ is set to $25<40$.  This tells us that the contradicting output indicates that the hyperparameters are wrong (although the inverse is not necessarily true).  After increasing the parameter of error margin, we obtained $L=20$ which should reflect the data.
> >
> > After that, we also tested our eigenvalue estimation algorithms by comparing to the exact DMD (dynamic mode decomposition).  For DMD, we use all the image data as vectors and assume that the dynamics is linear (which is not true) to compute the matrix.  Then, we tested our algorithm with the sample number $N=10^3,10^4,10^5$.  For $10^3,10^4$, only one eigenvalue $1+0i$ remained but for $10^5$ we obtained three sufficiently positive eigenvalues.
> > Those seem to be somewhat close to the three of four top eigenvalues of the DMD matrix.  From the results, we argue that, for some real applications where the system is not exactly linear, the eigenvalue estimation algorithm may not work well although it does not output *insane* values.  However, since the target system does not satisfy our assumptions, we cannot evaluate this result further.
> >
> > >Reduce the redundancy in the Introduction and Related Work, and, if possible, discuss how the techniques and results differ from the closest research work(s).
> >
> > Thank you for the suggestion; we would like to ask you to look at our response to your comment above.  We reduced the redundancy by making the introduction section more concise, added a motivation section summarizing our broad motivations behind our problem settings, and modified the related work section to better place our work in the relations to existing lines of work.  Our broad motivation is for reconstruction of dynamical system properties from partial and noisy observations.  For such a problem, we believe there exist two lines of work in the literature; (1) that of Takens and its extension, and (2) partially observable linear system identification and control.
> >
> > While the former studies more general nonlinear attractor dynamics from a sequence of scalar observations made by certain observation function having desirable properties, it is originally for the noiseless case.  Although there exist some attempts on extending it to noisy situations (such as (Martin Casdagli et al.)), provable estimation of some dynamic structure information with sample complexity guarantees (or nonasymptotic result) under fairly general noise case is not elaborated.  For the latter, the work typically consider additive Gaussian noise, controllable/observable systems, adversarial noise with fixed budget, the system with spectral radius strictly smaller than one etc. which are different sets of assumptions that those we have in this work.
> >
> > While we restrict ourselves to (nearly) periodic systems and bandit feedback scenarios, we do not assume control inputs or Gaussian noise etc.  In this regard, we are not proposing ''better'' algorithms than the existing work but are considering different problem setups that we believe are interesting and we hope the newly added motivation section further strengthens this perspective.

---

> > > ### Author Response · Authors · 2024-04-18
> > > **Additional responses**
> > >
> > > >Justify the quality and goodness of the results and guide parameter selection in practical applications (with reasonable justification).
> > >
> > > Thank you for the comment; for both goodness and parameter selections, we would like to ask you to look at the responses above.
> > >
> > > >Conduct some experiments on real-world data to mitigate the discrepancy between problem-motivating applications and the experiments in the paper.
> > >
> > > Thank you for the comment; we would like to ask you to look at our response above.

---

### Review · Reviewer_okMR · 2024-03-14

**Summary Of Contributions:**

This paper focuses on extracting structure from dynamical estimation problems with bandit feedback by use of exponential (Weyl) sums. These sums elicit spectral and periodic information about the system and allow for estimation of these quantities.  The authors then propose tractable and efficient algorithms for computing these quantities of interest, giving sample complexities and convergence results, and conclude with experiments.

**Audience:**

Yes

**Claims And Evidence:**

Yes

**Requested Changes:**

Examples: It would be nice to have a few examples explaining how the bandit feedback interacts with the linear dynamical system estimation.

Questions:
Is the framework of bandit feedback necessary for the results generated? Could we instead just assume that x_t is a parameter on which the reward hinges?
There are a few minor grammatical typos, e.g. Assumption 3.2 family should be families etc.

**Strengths And Weaknesses:**

Main Contributions:

Originality: This paper introduces a novel use of exponential sums to create asymptotic bounds of dynamical structure, i.e. period and spectrum of dynamic system. This mathematical machinery has not yet been applied to problems within the statistical learning framework, and this connection alone is a compelling research perspective.

Significance: The authors establish a litany of results, under the more loose assumptions of sub-Gaussian noise and with no claims of autonomy and observability, extending the previous dynamical systems literature into problems of bandit feedback.

Main Weaknesses:
Motivation: Although the mathematical complexity behind Weyl sums makes them a compelling topic of study, the motivation to use them within this context is lacking within the paper. An introduction to the related work about these sums as well as a section giving a brief introduction to their properties of note would be beneficial. In addition, motivation for studying the intersection of the two challenges faced in the work, dynamical system identification and bandit feedback would be beneficial.

Experiments/Algorithms :  The algorithms are novel, yet a discussion of the benefits and weaknesses and comparison to others in the literature is scarce.

---

> ### Author Response · Authors · 2024-04-18
> **Thank you so much for your constructive comments**
>
> We sincerely thank you for kindly taking your precious time reviewing our paper.
> We thank you for your very constructive comments that have helped us improve the clarity and motivations of our manuscript.
> Below, we answered to your questions and we revised the manuscript accordingly.
>
> >Motivation: Although the mathematical complexity behind Weyl sums makes them a compelling topic of study, the motivation to use them within this context is lacking within the paper. An introduction to the related work about these sums as well as a section giving a brief introduction to their properties of note would be beneficial.
>
> Thank you for the constructive comment; we added a brief ''paragraph'' in the introduction section that highlights the properties of the exponential sums used in this work with some illustrations.  Also, we added some sentences on how the exponential sums are studied in number theory community in the paragraph as well.  We put the added sentences here for reference: Exponential sums, also known as trigonometric sums, have developed as a significant area of study in number theory, employing various methods from analytical and algebraic number theory (see (Arkhipov et al. (2004)) for an overview). An exponential sum consists of a finite sum of complex numbers, each with an absolute value of one, and its absolute value can trivially be bounded by the number of terms in the sum.  However, due to the cancellation among terms, nontrivial upper and lower bounds can sometimes be established.
> Several classes of exponential sums with such nontrivial bounds are known, and in this study, we apply bounds from a class known as Weyl sums to extract information from dynamical systems using bandit feedback.  In the mathematical community, these bounds are valuable not only in themselves but also for applications in fields like analysis within mathematics (Katz 1990).
> This research represents an application of these bounds in the context of machine learning (statistical learning (estimation) problem) and is cast as one of the first attempts of opening the applications of number theoretic results to learning theory.
>
> >In addition, motivation for studying the intersection of the two challenges faced in the work, dynamical system identification and bandit feedback would be beneficial.
>
> Thank you for the constructive comment; we added a section stating the motivations from both theoretical (scientific) and practical perspectives with an illustration.  We discussed motivations of each aspect, namely (1) dynamic information recovery from noisy observation (please see the newly added illustration), (2) dynamic information recovery from scalar observations (in relation to Takens' theorem), (3) physical constant estimation (eigenvalue estimation), (4) dynamic mode decomposition (mode analysis), (5) attractor reconstruction, (6) communication capacity (when information is encoded by periodic signals), (7) bandit feedback representation of single-pixel camera, (8) bandit feedback applications and connections to reward maximization problems.
> Then we discussed the combinations of them, which is represented by a concrete example of the use of ''single-pixel camera'' (which is also often studied under the context of compressed sensing) for capturing the target that possibly follows (nearly) periodic (and linear) motion.  We added an illustration of this problem in the manuscript as well.  Bandit feedback in our work is a scalar observation made by a user-defined linear observation vector which is contaminated by noise, and it is general enough to capture practical examples including ''single-pixel camera'', channel estimation, and other examples such as those considered in reward maximization problems.

---

> > ### Author Response · Authors · 2024-04-18
> > **Additional responses**
> >
> > >The algorithms are novel, yet a discussion of the benefits and weaknesses and comparison to others in the literature is scarce.
> >
> > Thank you for the comment; to better place our work in the literature, we modified the related work section in addition to the motivation section that shows broader aspects.  Our broad motivation is for discussing reconstruction of dynamical system properties from partial and noisy observations.  For such a problem, we believe there exist two lines of work in the literature; (1) that of Takens and its extension, and (2) partially observable linear system identification and control.
> >
> > While the former studies more general nonlinear attractor dynamics from a sequence of scalar observations made by certain observation function having desirable properties, it is originally for the noiseless case.  Although there exist some attempts on extending it to noisy situations (such as (Martin Casdagli et al.)), provable estimation of some dynamic structure information with sample complexity guarantees (or nonasymptotic result) under fairly general noise case is not elaborated.  For the latter, the work typically consider additive Gaussian noise, controllable/observable systems, adversarial noise with fixed budget, the system with spectral radius strictly smaller than one etc. which are different sets of assumptions than those we have in this work (neither subset or superset).  While we restrict ourselves to (nearly) periodic systems and bandit feedback scenario, we do not assume control inputs or Gaussian noise etc.  In this regard, we are not proposing ''better'' algorithms than the existing work but are considering different problem setups that we believe are interesting and we hope the newly added motivation section further strengthens this perspective.
> >
> > To better clarify this relation, we added some illustration in the related work section as well.  In other words, as you mention, the weakness is, if any, the restrictions to nearly periodic dynamics and to bandit feedback scenarios.  We added the following sentence to the discussion section as well: Since we consider a rather novel problem setup, the setting itself would be seen as a limitation of our work....
> >
> > >Examples: It would be nice to have a few examples explaining how the bandit feedback interacts with the linear dynamical system estimation.
> >
> > Thank you for the comment; we would like you to look at our response to your comment regarding adding the motivation section as well.  One possible concrete example is the case of the use of ''single-pixel camera'' (which is also often studied under the context of compressed sensing) to capture the target that possibly follows (nearly) periodic (and linear) motion.  We added an illustration of this problem in the manuscript as well.
> >
> > >Questions: Is the framework of bandit feedback necessary for the results generated? Could we instead just assume that $x_t$ is a parameter on which the reward hinges?
> >
> > Thank you for the important comment: our analysis depends on the fact that the reward is observed through linear bandit feedback of the system parameter ($f^t(\theta)$) with a bound (Assumption 3).  As you suggest, for the nearly period estimation problem itself, it should be possible to generalize the problem setups and to even consider the sequence *not* generated by dynamics $f$ as long as some critical assumptions still hold.  However, as our broad motivation stems from the reconstruction of dynamic structure information from partially and noisy observations, and as bandit feedback possesses practical applicability, we would like to keep our presentation of our work in the current form.  (Thank you for your suggestion though)
> >
> > >There are a few minor grammatical typos, e.g. Assumption 3.2 family should be families etc.
> >
> > Thank you for your comments; we have gone through the manuscript and have fixed several typos.  For the part you mention, we fixed to ''... is an ascending family'' instead of ''... is a set of ascending ...''.

---

### Review · Reviewer_wo7U · 2024-04-04

**Summary Of Contributions:**

This paper considers the following dynamical system:
$r_t(x_t)= {f^t(\theta)}^\top x_t + \eta_t,$.
The goals of the paper are:

1. Estimate the length of the period from the collection of rewards with sample efficiency guarantee.
2. Estimate eigenvalues of a linear dynamical system with sample efficiency guarantee.

**Audience:**

Yes

**Claims And Evidence:**

Yes

**Requested Changes:**

It would be great if the authors could clearly delineate the technical novelties required to deal with the issues mentioned in the Weakness section.

**Strengths And Weaknesses:**

**Strength**

The paper successfully matches the claims that it makes. It does extend the literature on period estimation and eigenvalue estimation of dynamical system to the sub-Gaussian setting (Theore 4.3 and Theorem 4.8). I like the use of Weyl's sum to establish the guarantees.

**Weakness**

I have some reservations regarding the novelty of the paper.

Extending the Gaussian results to the sub-Gaussian is pretty straightforward. It just requires a simple use of Azuma-Hoeffding inequality and Lemma D.1. Lemma D.1 and its variations have been used across literature.

Although the use of Weyl's sum technique is new, the main result follows very simply from the existing bounds on Weyl's sum, e.g., Lemma 4.6.

---

> ### Author Response · Authors · 2024-04-18
> **Thank you so much for your comments**
>
> We sincerely thank you for kindly taking your precious time reviewing our paper.  Your comments have helped us further clarify our ideas and motivations.
> Below, we answered to your questions and we revised the manuscript accordingly.
>
> >Extending the Gaussian results to the sub-Gaussian is pretty straightforward. It just requires a simple use of Azuma-Hoeffding inequality and Lemma D.1. Lemma D.1 and its variations have been used across literature.
>
> Thank you for the comment; indeed, we typically use Azuma-Hoeffding inequality to deal with sub-Gaussian case, and it is standard.  And our work starts with this fact that a general form of noise such as that drawn from sub-Gaussian distributions requires us to use concentration of measures.  We are not at all claiming that the use of concentrations of measure is novel.  Our theoretical motivation stems from the dilemma that this ''averaging'' procedure may erase some important dynamic structural information; to strengthen this argument, we added the motivation section on the use of concentration of measures for dynamic environment in addition to other motivations including reconstruction of dynamical system information and some practical motivations.  As an example, please consider two different dynamics, namely a fixed-point attractor dynamics converging to the fixed point $(0,0)$ and a limit-cycle attractor dynamics converging to the circular motion of radius one with the center $(0,0)$.  When averaging out the states over a sufficiently long time horizon, both will produce the outputs that are close to $(0,0)$.  In this case, those two dynamical systems are no longer distinguished clearly.  This example is rather for an illustrative purpose, but our research question is basically about what information we can provably extract from noisy observations under dynamic environments.  We emphasize again that our manuscript is *not* claiming that we are proposing novel concentration inequality but is motivated by the fact that the use of concentration of measure may cause some issues for dynamic environments.
>
> >Although the use of Weyl's sum technique is new, the main result follows very simply from the existing bounds on Weyl's sum, e.g., Lemma 4.6.
>
> Thank you for the comment; it would be relatively simple once we know the exact problem caused by the concentration of measure for dynamic environments and that we know that the asymptotic results on the exponential sums could be used to remedy this problem.  Although our work carefully combines linear algebraic manipulations, the exponential sums (*with a novel extension to matrix sums*), and concentration of measures, each step of the math is not that complicated process.  However, we believe that the benefits of our work in the community and their potential for further applications are nontrivial in the sense that (1) we introduced a novel use of the exponential sums and opened up a way to partially bridge statistical learning theory and (analytic) number theory, (2) defined several mathematical concepts such as *nearly period*, *aliquot nearly period*, and $(\theta,a)$-*distinct eigenvalues*, and (3) identifying provably obtainable dynamic structural information, which will potentially be of interests in the applications we listed in the motivation section.  We believe simpleness of the proof itself does not harm the importance of the work.
>
> >It would be great if the authors could clearly delineate the technical novelties required to deal with the issues mentioned in the Weakness section.
>
> Thank you for the comment; we assume you ask us to sketch the novelties in our work.  We ask you to please read the responses above where we remark that the novelties are highlighted by the use of the exponential sum techniques in the statistical estimation problems and by the careful combination of the results of linear algebra, exponential sums (e.g., the Weyl sum is extended to the form of matrix sums) and statistical methods.
> We hope our motivation and message have been conveyed correctly in this respect.

---

### Author Response · Authors · 2024-04-18
**To all of the reviewers**

Thank you so much for your time reviewing our work; your comments and suggestions have helped us clarify and improve our manuscript.
We responded to each of your comment and revised our manuscript.

---

### Decision · Action_Editor_VSuk · 2024-06-13

**Recommendation:** Accept with minor revision

**Comment:**

Reviewers found that the results in this paper follow immediately from existing works. However, novelty is not a criteria in TMLR, so I will argue that there is an audience for this paper.

I suggest authors to go over the reviews carefully and make modifications that would address their concerns as much as possible.

**Audience:**

I am not certain if the content of this paper is rich enough to generate interest among the TMLR audience.

**Claims And Evidence:**

Claims and proofs in this paper appear to be correct. Though they are straightforward extensions of existing results.

---

> ### Author Response · Authors · 2024-06-28
> **Thank you very much for your handling our paper**
>
> Thank you very much for your commitment to the process of our submission and to the TMLR community.  We will go over the reviews again carefully to improve some parts of the paper, and will update the paper by the specified time.
>
> Sincerely, On behalf of the authors

---

> ### Author Response · Authors · 2024-07-20
> **Thank you; we have updated the paper and uploaded the camera-ready version as instructed**
>
> Thank you so much again for your handling our paper.
> Thank you for your comments, and thank you for your fair decision and views on our work.
> We went through the reviews again and improved some parts of the paper, and uploaded a camera-ready version as instructed.
>
> Here we list the updates we have made in the paper for your reference.
> we (1) added a paragraph of summary of technical contributions and challenges in introduction (2) added slightly more texts to the motivation and caption of Figure 4, and added slightly more discussions on hyperparameter choice (in experiment and discussion section) to further clarify some aspects of our work, and (3) added the following text in introduction to further close the gap between the stated motivation and the results: "specifically, we mainly study theoretical aspects of recovering structural information under a novel set of model assumptions."